# OneFlow: Concurrent Mixed-Modal and Interleaved Generation with Edit Flows

## Abstract

We present **OneFlow**, the first non-autoregressive multimodal model that enables variable-length and concurrent mixed-modal generation. Unlike autoregressive models that enforce rigid causal ordering between text and image generation, One-Flow combines an insertion-based Edit Flow for discrete text tokens with Flow Matching for image latents. OneFlow enables concurrent text-image synthesis with hierarchical sampling that prioritizes content over grammar. Through controlled experiments across model sizes from 1B to 8B, we demonstrate that One-Flow outperforms autoregressive baselines on both generation and understanding tasks while using up to 50% fewer training FLOPs. OneFlow surpasses both autoregressive and diffusion-based approaches while unlocking new capabilities for concurrent generation, iterative refinement, and natural reasoning-like generation.

## 1 Introduction

Native Multimodal Models — models capable of handling both multimodal understanding and generation within a single backbone — have advanced considerably in visual understanding and generation. These models typically employ a unified transformer architecture with next-token prediction to handle both discrete and continuous generation (Team, 2024; Wu et al., 2025; Ma et al., 2025; Deng et al., 2025; Zhou et al., 2025). Recent work like Transfusion (Zhou et al., 2025) and Show-O (Xie et al., 2024) demonstrates that leveraging modality-specific training objectives within shared architectures can significantly improve performance, particularly on continuous modalities such as vision.

However, both autoregressive (AR) and diffusion-based multimodal approaches face fundamental architectural constraints. Autoregressive models can handle interleaved data but require strict sequential generation — each image must be fully completed before text generation can continue, preventing simultaneous cross-modal refinement. Conversely, diffusion-based multimodal models such as MMaDA (Yang et al., 2025), FUDOKI (Wang et al., 2025), and Unidisc (Swerdlow et al., 2025) enable simultaneous mixed-modal generation but only for predetermined single text-image pairs where modality assignments must be known a priori and rely on independent time schedules for each modality. Neither paradigm supports the simultaneous generation of variable-length interleaved sequences.

We present **OneFlow**, the first model to achieve simultaneous generation of interleaved data. Unlike autoregressive models that enforce sequential completion of each modality, and unlike diffusion models restricted to fixed length generation, OneFlow combines an insertion-based discrete text generation using Edit Flows with Flow Matching for image generation. This enables concurrent refinement of both text and images with per-image time schedules, using a novel interleaved time schedule.

Through controlled experiments across various model sizes and compute regimes, we demonstrate that OneFlow outperforms both autoregressive (AR) and diffusion baselines on generation and understanding tasks while requiring 50% fewer training FLOPs. Moreover, we find that concurrent mixed modal pretraining yields 4% relative improvement on VQA and 1.5% on image generation over sequential pretraining. We summarize our contributions below.

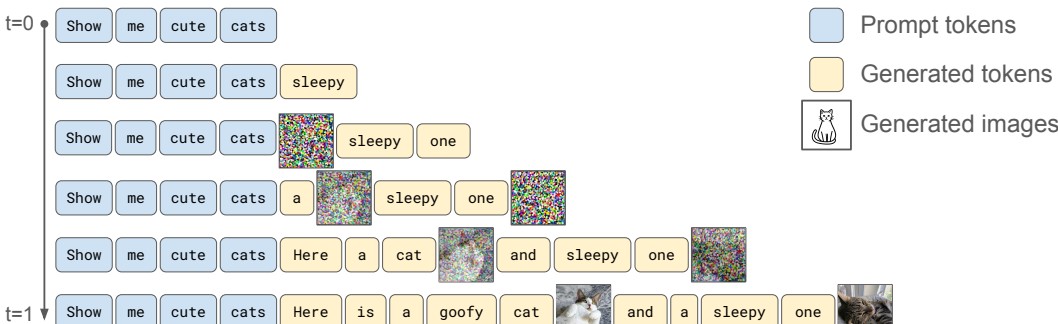

Figure 1: OneFlow is a variable-length non-autoregressive model that can concurrently generate interleaved text and variable number of images using insertions as a primitive operation.

**Contributions:**

1. We introduce OneFlow, a non-autoregressive multimodal model that unifies image and text generation under a simultaneous Edit Flow and Flow Matching framework.

2. OneFlow outperforms or is competitive with existing AR and diffusion-based models on a diverse range of image generation and image understanding benchmarks while unlocking new capabilities such as concurrent generation.

3. OneFlow's mixed-modal training boosts performance over uni-modal generation across a wide range of benchmarks.

4. Through controlled experiments from 1B to 8B, we find that OneFlow scales better than autoregressive multimodal models, particularly with mixed-modal training.

## 2 ONEFLOW: MIXED-MODAL GENERATION THROUGH FLOW MATCHING

OneFlow handles multiple modalities through a sequence model, where elements in the sequence can be discrete tokens or continuous embeddings, *e.g.*, of images. Concretely, let $\mathcal{T}$ denote the space of a single element of the sequence, which can take either a discrete value, up to some fixed vocabulary size $M$, or a continuous value, *i.e.*, $\mathcal{T} = [M] \cup \mathbb{R}$. Then our state space is defined as the set of all possible sequences up to some maximum length $N$, *i.e.*, $\mathcal{X} = \bigcup_{n=0}^{N} \mathcal{T}^n$.

During generation, our model transforms noisy sequences into clean sequences. We do this by combining discrete-valued and continuous-valued generative processes. Specifically, we make use of the Edit Flows (Havasi et al., 2025) framework which enables variable-length sequence generation through the use of edit operations. It starts with a noisy sequence and iteratively applies edits until it is denoised into a generation. We focus on the *insertion* capabilities of Edit Flows, which is conceptually simple yet extremely powerful, as it allows inserting arbitrary number of tokens—and images—into the generated sequence. When images are inserted, we initialize them with noise and then use Flow Matching (Lipman et al., 2024) to generate the image. Since the same model predicts both the text edits and the image denoising, OneFlow achieves variable-length, non-autoregressive joint image and text generation. In the following, we state equations with only intuitive justifications and explanations. Full mathematical details and derivations can be found in Appendix B.

### 2.1 DISCRETE TEXT GENERATION VIA EDIT FLOWS

Edit Flows uses a continuous-time Markov chain (CTMC) to iteratively refine variable-length discrete sequences. We start with an empty sequence $X_0 = \emptyset$ at time $t = 0$, and transform the sequence through insertion operations. Let $\text{ins}(x, i, a)$, $x \in \mathcal{X}, i \in \{1, \ldots, n\}, a \in \mathcal{T}$, be the sequence resulting from inserting the token value $a$ to the right of position $i$ of the sequence $x$, resulting in

$$\text{ins}(x, i, a) = (x^1, \ldots, x^i, a, x^{i+1}, \ldots, x^n). \tag{1}$$

This forms the primitive operation that we use during generation.

During training, we take a data sequence $X_1$ and randomly delete tokens with equal probability to obtain $X_t$. This defines the process $X_{[0,1]}$ that we will fit to. The probability of each token being deleted is set by a monotonic scheduler $\kappa_t$ with $\kappa_0 = 0, \kappa_1 = 1$.

$$\mathbb{P}(x^i \text{ in } X_t) = \kappa_t, \qquad \text{for each } x^i \in X_1. \tag{2}$$

In preliminary experiments, we tested different $\kappa_t$ but found that the linear schedule $\kappa_t = t$ works most consistently across our diverse benchmarks. *Deleted tokens are removed from the sequence.* Noting that on average we retain $\mathbb{E}_t[\kappa_t]$ fraction of the original tokens, with the linear schedule we retain 50% of the data sequence. This can lead to significant FLOPs savings during training, and tuning the scheduler can save even more if desired.

**Parameterization.** The parameterization of an Edit Flow model for insertions naturally decomposes into two predictions: (i) how many tokens are missing at the right of position $i$, and (ii) which tokens are missing. Thus, at each position $i$ of the sequence, our model outputs two quantities

- $\lambda^i : \mathcal{X} \to \mathbb{R}^+$ is a scalar that predicts the *number of missing tokens* between $i$ and $i+1$.
- $Q^i : \mathcal{X} \to [M]$ is a normalized distribution that predicts *what tokens are missing*.

These two predictions form the CTMC rate and gives the transitions (up to $o(h)$ error),

$$\mathbb{P}\left(X_{t+h} = \text{ins}(X_t, i, a) \mid X_t\right) = h \frac{\dot{\kappa}_t}{1-\kappa_t} \lambda^i(X_t) Q^i(a|X_t). \tag{3}$$

The ratio $\frac{\dot{\kappa}_t}{1-\kappa_t}$ dictates the distribution of insertion times according to the schedule $\kappa_t$ imposed during training (2), where $\dot{\kappa}_t = \frac{d\kappa_t}{dt}$. Note that unlike prior work (Havasi et al., 2025), we factor out this ratio $\frac{\dot{\kappa}_t}{1-\kappa_t}$ from the rate predictions and use a simplified model that is independent of $t$. Practically, we do not feed time values into the network for predicting insertions. While not theoretically justified, we found this $t$-independence assumption to work better in practice, likely because $X_t$ already contains sufficient information for predicting the insertions. We report the ablation on $t$-independence in Appendix F.4.

**Insertion prediction** ($\lambda^i$)**.** The main component that determines whether insertions occur is the prediction head $\lambda^i$, which is trained by regressing onto the number of missing tokens. Each position $i$ of the noisy sequence $X_t$ has a corresponding number of missing tokens $k^i$, which is the number of deleted tokens between $X_t^i$ and $X_t^{i+1}$. The original Edit Flows loss was constructed through a choice of Bregman divergence (Holderrieth et al., 2024) which results in

$$\ell_{\text{Poisson}}(\lambda^i) = \sum_i \lambda^i(X_t) - k^i \log \lambda^i(X_t). \tag{4}$$

Alternatively, (4) can be interpreted as the negative log-likelihood of a Poisson distribution, so $\lambda^i$ is trained to fit a Poisson distribution to model missing token counts $k^i$. However, the distribution of $k$ has a very high concentration around zero missing count. Furthermore, during sampling, the key prediction is whether the missing token count is zero or nonzero. As such, we explicitly model the probability of inserting zero tokens.

$$\mathbb{P}(k=0) = \pi, \quad \text{and} \quad \mathbb{P}(k) = (1-\pi)\text{Pois}(k; \lambda_{\text{nonzero}} \mid k > 0) \quad \text{for } k > 0 \tag{5}$$

where $\pi \in (0,1)$ is the probability of observing zero, and $\lambda_{\text{nonzero}} > 0$ is the rate parameter but restricted to only modeling the distribution of nonzero counts. We train $\pi$ by using a binary cross entropy (BCE) loss to detect if the missing count is zero, and we train $\lambda_{\text{nonzero}}$ using the original loss ((4)) on nonzero counts. For sampling (3), we can use the expectation $\lambda^i(X_t) = (1 - \pi^i(X_t))\lambda_{\text{nonzero}}^i(X_t)$. However, we found that a consistently better sampling strategy is to first sample whether there are zero insertions using $\pi$, then simply use the rate $\lambda_{\text{nonzero}}^i(X_t)$ if there are nonzero insertions.

**Bag-of-tokens prediction** ($Q^i$)**.** To determine what token to insert at each position, we make use of the output head $Q$ which is a softmax over the discrete vocabulary $[M]$. We use the same Edit Flows loss, which is a sum of cross-entropy loss. Let $\mathcal{A}_i$ denote the set of deleted tokens between $X_t^i$ and $X_t^{i+1}$, then for each position $i$, the loss is

$$\ell_{\text{tokens}}(Q^i) = -\sum_{a \in \mathcal{A}_i} \log Q^i(a|X_t). \tag{6}$$

**Combined loss.** At each training iteration, we randomly delete tokens from the data sequence, and learn to predict the set of missing tokens at each position, resulting in the total insertion loss:

$$\mathcal{L}_{\text{text}} = \mathbb{E}_{t, X_t | X_1} \left[ \frac{1}{n} \sum_{i=1}^{n} \ell_{\text{tokens}}(Q^i) + \ell_{\text{Poisson}}(\lambda_{\text{nonzero}}^i) \mathbf{1}_{[k_i > 0]} + \ell_{\text{BCE}}(\pi^i) \right] \tag{7}$$

where $n$ is the length of the noisy sequence $X_t$. Note that this differs from the original training objective in Edit Flows (Havasi et al., 2025) which additionally weights the loss by the factor $\frac{\dot{\kappa}_t}{1 - \kappa_t}$, not affecting the optimal solution. We found that not using this factor produces better results.

## 2.2 Continuous image generation via Flow Matching

Following standard practice, we generate images starting from a Gaussian noise of fixed dimension $N_{\text{img}}$, applying a deterministic generation procedure that follows an ordinary differential equation. Let $Y_t \in \mathbb{R}^{N_{\text{img}}}$ denote the noisy image, then the generative process is

$$\frac{d}{dt} Y_t = v(Y_t, t), \qquad Y_0 \sim \mathcal{N}(0, I), \tag{8}$$

where $v : \mathbb{R}^{N_{\text{img}}} \times \mathbb{R} \to \mathbb{R}^{N_{\text{img}}}$ is a velocity field that determines the direction to transform $Y_t$ into a clean sample by $t = 1$. During training, we sample a noise $Y_0$ and obtain $Y_t$ with a linear schedule $Y_t = tY_1 + (1 - t)Y_0$. The Flow Matching loss can then be written as

$$\mathcal{L}_{\text{image}} = \mathbb{E}_{t, Y_0, Y_1} \| v(Y_t, t) - (Y_1 - Y_0) \|^2. \tag{9}$$

In OneFlow, we use a pretrained autoencoder to map images into latent space. We then design the velocity network $v(\cdot)$ to use a shared Transformer backbone as text but with additional U-Nets to downsample and upsample between the backbone and autoencoder embedding spaces, making use of the same architectural design as Transfusion (Zhou et al., 2025). See illustration in Figure 13.

## 2.3 Concurrent Mixed-modal Generation

To generate multiple modalities, we simply concatenate them into a single sequence. We now present two multimodal time schedules, an independent schedule that can be used when the number of images is known, and an interleaved schedule that needs to be used when the number of images is arbitrary. OneFlow is designed to work with variable-length text and variable number of images.

**Independent mixed-modal generation.** We can consider the simple case with a fixed number of images—typically one. In such case, we can generate both the text and image simultaneously by using two time values $t_{\text{text}}$ and $t_{\text{img}}$, where $t_{\text{text}}$ determines the state of the insertion generation process and $t_{\text{img}}$ determines the image generation process. Following prior work, we simply set independent time schedules, one for the text and one for each image. This allows the modalities to be concurrently generated and be dependent on each other during the generation process. However, this naïve process does not allow us to insert images.

**Interleaved mixed-modal generation.** A much more complicated setting arises when the number of images is variable and images are being inserted as part of the generation process. Similar to the text-only setting, we start generating from the empty sequence. We then model image insertion as a special token value `<|image|>`, which is added to the token prediction output $Q$. During generation, when the model predicts an image insertion, we insert noise embeddings of dimension $N_{\text{img}}$ into the sequence to represent an inserted image initialized at $t_{\text{img}} = 0$.

$$\text{ins}(x, i, \texttt{<|image|>}) = \left( x^1, \ldots, x^i, x_{\text{img}}^1, \ldots, x_{\text{img}}^{N_{\text{img}}}, x^{i+1}, \ldots, x^n \right), \qquad x_{\text{img}}^i \sim \mathcal{N}(0, I). \tag{10}$$

Subsequent steps during generation would then simultaneously generate the image embeddings while also performing more insertions into the sequence. However, since the image is generated at a later time, this implies there is a delay between the image time and the text time, *i.e.* $t_{\text{img}} \leq t_{\text{text}}$, which needs to be taken into account during training.

During training, we need to ensure that the text and image noise levels are consistent with with the ones seen during generation. Based on the schedule in (2), the time at which an insertion happens is a random variable that has $\kappa$ as its cumulative density function, so the time difference between the inserted image time $t_{\text{img}}$ and the initial text time $t_{\text{text}}$ is given by

$$t_{\text{img}} = t_{\text{text}} - \kappa^{-1}(u), \qquad \text{where } u \sim \text{Unif}(0, 1). \tag{11}$$

| Model | Size | Text | Image | Image Generation | | | | Captioning | | |
|---|---|---|---|---|---|---|---|---|---|---|
| | | | | FID↓ | CLIP↑ | DPG↑ | WISE (c.) ↑ | CIDEr↑ | ROUGE↑ | BLEU4↑ |
| *Unified MLM* | | | | | | | | | | |
| MetaMorph (Tong et al., 2024b) | 7B | AR | AR | 11.8 | 26.6 | – | – | – | – | – |
| LMFusion (Shi et al., 2024b) | 7B | AR | Diff | 14.0 | 24.4 | – | – | 38.4 | – | – |
| Transfusion (Zhou et al., 2025) | 7B | AR | Diff | 16.0 | 26.5 | 77.8 | – | 33.7 | – | – |
| Janus-Pro (Chen et al., 2025) | 1.5B | AR | AR | 15.2$^\dagger$ | 26.0$^\dagger$ | 82.0$^\dagger$ | 0.20 | – | – | – |
| Janus-Flow (Ma et al., 2025) | 1.5B | AR | FM | 12.4$^\dagger$ | 26.1$^\dagger$ | 80.1$^\dagger$ | 0.13 | – | – | – |
| Bagel (Deng et al., 2025) | 7B | AR | FM | 27.7$^\dagger$ | 26.2$^\dagger$ | 84.7$^\dagger$ | 0.44 | – | – | – |
| *Multimodal Diffusion* | | | | | | | | | | |
| UniDisc (Swerdlow et al., 2025) | 1.4B | Mask | Mask | 23.9 | – | – | – | – | – | – |
| D-DiT (Li et al., 2025) | 2B | Mask | Diff | – | – | – | – | 56.2 | – | – |
| Muddit (Shi et al., 2025) | 1B | Mask | Mask | – | – | – | – | 59.7 | – | – |
| MMaDA (Yang et al., 2025) | 8B | Mask | Mask | 33.2$^\dagger$ | 25.1$^\dagger$ | 74.2$^\dagger$ | 0.67 | – | – | – |
| FUDOKI (Wang et al., 2025) | 1.5B | DFM | DFM | – | – | 83.6 | – | – | – | – |
| *Controlled Comparisons* | | | | | | | | | | |
| AR + FM Ablation | 1B | AR | FM | 12.2 | 26.5 | 73.4 | 0.61 | 123.9 | 57.2 | 0.39 |
| Mask + FM Ablation | 1B | Mask | FM | 11.3 | 26.5 | 75.5 | 0.64 | 128.4 | 58.6 | 0.39 |
| **OneFlow** | 1B | EF | FM | 12.1 | 26.6 | 79.1 | 0.62 | 138.1 | 60.8 | 0.41 |
| **OneFlow Mixed** | 1B | EF | FM | 9.7 | 26.6 | 80.3 | 0.63 | 139.8 | 60.9 | 0.42 |
| **OneFlow** | 8B | EF | FM | 10.7 | 26.7 | 79.3 | 0.65 | 141.1 | 61.1 | 0.42 |
| **OneFlow Mixed** | 8B | EF | FM | 9.5 | 26.6 | 80.4 | 0.68 | 142.1 | 61.1 | 0.43 |

Table 1: **Image generation and captioning benchmarks after multimodal pretraining.** We find that mixed-modal training consistently improves performance. $^\dagger$Evaluated using official open-source model weights. Highlighting denotes best results across all models.

We call this the *interleaved time schedule*, which imposes a distributional dependency between the time values $t_{\text{img}}$ for each image and the text time $t_{\text{text}}$. In order for the model to learn to fully generate all images, during training we sample from an extended time interval, $\tau_{\text{text}}$ from $[0, 2]$, since the <|image|>token can be inserted at $\tau_{\text{text}} = 1.0$ at the latest, and fully denoised by $\tau_{\text{text}} = 2.0$. The probability for each token being in $X_t$ is then determined by $\kappa(\min\{1, \tau_{\text{text}}\})$ in place of (2). We also sample for each image an extended time value $\tau_{\text{img}} = \tau_{\text{text}} - \kappa^{-1}(u)$. Finally, we determine if an image is deleted from the sequence by checking $\tau_{\text{img}} < 0$, and if so, the insertion loss ((7)) will include the <|image|> token which the model would learn to insert. Otherwise if $\tau_{\text{img}} \geq 0$, the image is in the sequence and we set $t_{\text{img}} = \min\{1, \tau_{\text{img}}\}$; using the Flow Matching loss (9) to train the velocity. A detailed derivation and more in-depth explanation can be found in Appendix B.1.

## 3 EXPERIMENTS

To study the performance of OneFlow against AR, we design a series of controlled experiments. We establish strong baselines by comparing both our approach and AR against other unified multimodal models in the literature. Finally, we explore the novel capabilities that OneFlow enables beyond existing methods. We present our experimental results through five research questions:

§3.1    How does OneFlow scale compared to AR?
§3.2    What is the impact of mixed-modal vs. sequential pretraining?
§3.3    What emergent behaviors does OneFlow exhibit during generation?
§3.4    How does OneFlow compare to other unified multimodal models?
§3.5    What new capabilities does OneFlow enable beyond existing methods?

**Training stages.** Our training consists of two main stages: multimodal pretraining and instruction finetuning. During the pretraining stage, we use a mixture of image understanding and image generation data to learn representations for both image and text. We trained with a sequence length of 512 and a global batch size of 4096. We can set the mixed generation probability (the likelihood of concurrently generating clean text and images from a noisy input) to be either 0 or 0.2.

For finetuning, we use a mixture of VQA, text, and interleaved data to give the model the ability to respond to visual question answering problems. We also fine-tune on image generation data at a higher resolution of $512\times512$ to improve the model's image generation capabilities. We study the model's behavior at the 1B scale for our ablations and controlled experiments, and the scaling trend up to 8B is detailed in Section 3.1.

| | | General | | | | Knowledge | | OCR & Chart | | | Vision | Halluc. |
|---|---|---|---|---|---|---|---|---|---|---|---|---|
| Model | Params | MMB | VQAv2 | GQA | MME | MMMU | AI2D | DocVQA | ChartQA | TextVQA | RealWorld | POPE |
| *Multimodal LM* | | | | | | | | | | | | |
| Show-O (Xie et al., 2024) | 1.3B | – | – | 61.0 | 1232.9 | 27.4 | – | – | – | – | – | 84.5 |
| MetaMorph (Tong et al., 2024b) | 7B | 75.2 | – | – | – | 41.8 | – | – | 37.1 | 60.5 | 58.3 | – |
| Janus-Flow (Ma et al., 2025) | 1.5B | 74.9 | 79.8 | 60.3 | 1333.1 | 29.3 | – | 64.6 | 55.5 | – | – | 88.0 |
| Janus-Pro† (Chen et al., 2025) | 1.5B | 73.4 | 67.9 | 59.3 | 1443.0 | 33.4 | 62.8 | 21.2 | 35.8 | 53.9 | 53.5 | 84.8 |
| Janus-Pro† (Chen et al., 2025) | 7B | 76.9 | 74.1 | 62.0 | 1531.0 | 38.2 | 68.1 | 24.3 | – | 57.2 | 56.4 | 85.2 |
| BAGEL (Deng et al., 2025) | 1.5B | 79.2 | – | – | 1610.0 | 43.2 | – | – | – | – | – | – |
| BAGEL (Deng et al., 2025) | 7B | 85.0 | – | – | 1687.0 | 55.3 | – | – | – | – | – | – |
| Mogao (Liao et al., 2025) | 7B | 75.0 | – | 74.6 | 1592.0 | 60.9 | – | – | – | – | – | 88.9 |
| *Mask Diffusion* | | | | | | | | | | | | |
| Muddit (Shi et al., 2025) | 1B | – | 67.7 | 57.1 | 1104.6 | – | – | – | – | – | – | – |
| D-DiT (Li et al., 2025) | 2B | – | 60.1 | 59.2 | 1124.7 | – | – | – | – | – | – | 84.0 |
| MMaDA (Yang et al., 2025) | 8B | 68.5 | 76.7 | 61.3 | – | 30.2 | – | – | – | – | – | 86.1 |
| *Discrete Flow* | | | | | | | | | | | | |
| FUDOKI (Wang et al., 2025) | 1.5B | 73.9 | – | 57.6 | 1485.4 | 34.3 | – | – | – | – | – | 86.1 |
| *Controlled Comparisons* | | | | | | | | | | | | |
| AR + FM Ablation | 1B | 64.7 | 66.0 | 55.0 | 1394.8 | 28.9 | 59.1 | 22.7 | 35.5 | 48.3 | 50.5 | 85.6 |
| Mask + FM Ablation | 1B | 66.0 | 64.4 | 55.6 | 1462.2 | 28.4 | 55.1 | 18.7 | 34.7 | 44.6 | 50.6 | 84.6 |
| **OneFlow** | 1B | 69.0 | 67.7 | 57.8 | 1497.1 | 29.8 | 58.5 | 23.8 | 35.0 | 50.4 | 50.6 | 85.7 |
| **OneFlow** | 8B | 72.5 | 73.7 | 61.9 | 1542.5 | 33.1 | 63.4 | 37.1 | 42.1 | 58.6 | 54.0 | 86.3 |

Table 2: **VQA performance comparison.** OneFlow outperforms AR and Mask models across all benchmarks in controlled experiments using identical finetuning data. Highlighting shows best results in the 1B controlled comparisons. Our results are also competitive with existing autoregressive and discrete diffusion models. †Evaluated using official open source weights.

**Datasets.** For multimodal pretraining, we use image-text pairs from a filtered version of the Conceptual Captions dataset (CC12M (Sharma et al., 2018)), the YFCC dataset (Thomee et al., 2016), and licensed data, for a total of 400M examples. During instruction finetuning, we use a filtered image portion of the PerceptionLM dataset (Cho et al., 2025), interleaving data from Chameleon (Team, 2024), and Cambrian-7M (Tong et al., 2024a) dataset.

**Baselines.** To evaluate our model's performance against existing methods, we compare against two baselines: (1) an autoregressive (AR) + Flow Matching (FM) multimodal model based on Transfusion (Zhou et al., 2025), where text tokens are generated autoregressively and image tokens via FM, and (2) a masked diffusion model based on LLaDA (Nie et al., 2025). For the masked diffusion baseline, we tested two sampling variants: low-confidence and random remasking, with random remasking performing better across all experiments. Unlike Transfusion, we follow Janus-Flow (Ma et al., 2025) and adopt a dual-encoder setup. For image encoders, we use a pretrained SigLIP2 ViT-SO400M-16@512 (Tschannen et al., 2025) for understanding and an SD3 VAE (Esser et al., 2024) for generation. Following Transfusion, we use U-Net adapters.

**Evaluation setup.** Following Cambrian (Tong et al., 2024a) and PLM (Cho et al., 2025), we group VQA tasks into five groups: General, Knowledge, OCR & Chart, Hard Perception, and Hallucination. We evaluate image generation quality using FID (Heusel et al., 2017) on COCO-2014 (Lin et al., 2014) validation set. To assess prompt alignment, we report CLIPScore (Hessel et al., 2021) and DPG-Bench (Hu et al., 2024). Additionally, we include WISE (Niu et al., 2025) cultural to better understand knowledge-based generation.

## 3.1 ONEFLOW SCALES BETTER THAN AR

In this experiment, we study the performance of OneFlow and AR in controlled settings at various model sizes and token counts. To ensure OneFlow has no advantage in data-constrained settings, we trained both models on 2B image-text pairs over 500k iterations using a batch size of 4096. Both models were initialized from Llama 3.2 1B (AI@Meta, 2024). For AR, the number of tokens

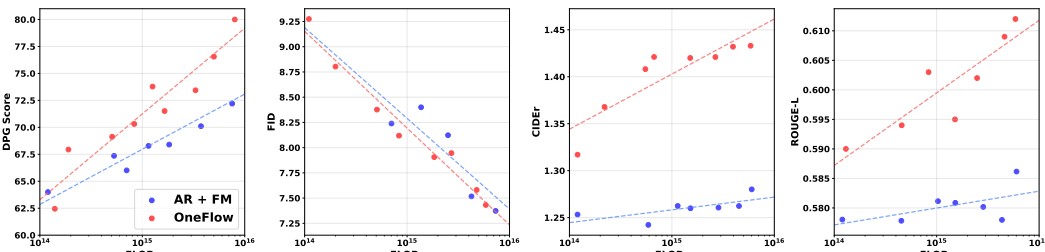

Figure 2: **Performance of OneFlow vs. AR baseline models at different model scales, data and compute.** For text-to-image generation, we report DPG-Bench and FID. For image-to-text caption quality, we report CIDEr and ROUGE. In every benchmark, OneFlow consistently exhibits better scaling laws than AR.

predicted during training equals the sequence length, whereas for OneFlow, the number of predicted tokens corresponds to the number of deleted tokens, which on average is 50% of the data sequence.

We find that OneFlow scales better than AR on every benchmark. This scaling advantage is especially pronounced on DPG Bench, where OneFlow scales significantly better than AR. Conversely, for image captioning, OneFlow shows a notable performance gap relative to AR. Figure 2 visualizes the scaling trend, and the final metrics after training are shown in Table 1, along with a comparison against other state-of-the-art models. We provide qualitative examples of where OneFlow outperforms AR in Appendix F.7.

## 3.2 Mixed Modal Pre-training Enables Better Generation and Understanding

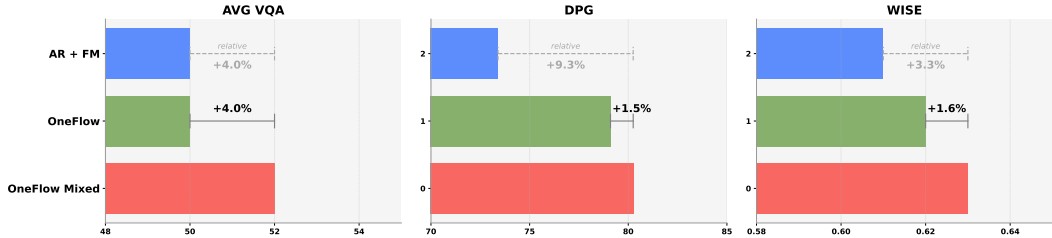

Figure 3: **Mixed modal vs Sequential pretraining.** Mixed modal pretraining vs sequential pretraining. Mixed pretraining achieves 4% relative improvement on VQA tasks and slight improvements on image generation as well.

In this section, we study the impact of mixed modal pretraining. We investigate whether concurrent mixed modal pretraining and sequential pretraining affect downstream understanding tasks. We train two 1B variants: one using sequential pretraining (T2I or I2T), and another with the same data but where 20% of examples use concurrent generation.

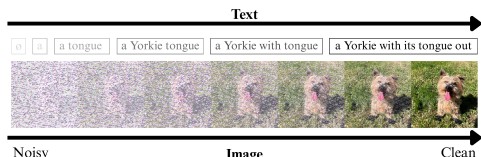

Figure 4: Mixed modal concurrent training.

Figure 3 indicates that when starting from a concurrent model, downstream image understanding improves by 5% relatively. These results demonstrate the impact of concurrent mixed modal pre-training dense prompt alignment. We also finetune the pretrained model from Section 3.1 on a mixture of interleaving data, filtered PLM data, and text data to evaluate downstream performance. We report the ablation on $k$-scheduler in Appendix F.5.

## 3.3 Hierarchical Generation Enables Reasoning

In Figure 5, we present OneFlow's sampling process when prompted with a visual question. In response to prompts such as *" {question} Explain why."*, OneFlow generates a reasoning chain before arriving at the final answer, without any Chain-of-Thought (CoT) (Wei et al., 2022) prompting or RL post-training. For example, when asked *"Is there a snowboard in the image? Explain why."*, the model first implicitly performs visual search by examining the image and searching through

**Question:** Is there a snowboard in the image? Explain your answer.

Yes, there is a snowboard in the image. This is evident from the visible snowboard that the snowboarder is riding on the slope. The snowboarder is positioned on the snowboard, making it the central focus of the image.

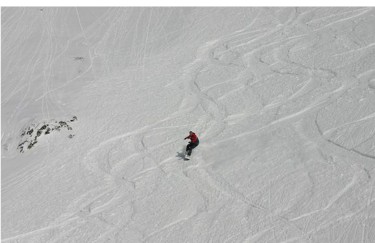

**Sampling Step**

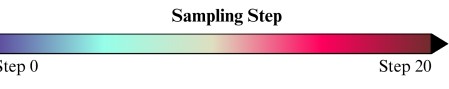

Step 0                                   Step 20

**Question:** Subtract all large rubber spheres. Subtract all big shiny cylinders. How many objects are left? Explain your answer.

In the image, there is one object that matches the description of the large sphere: the green sphere, and the only large shiny cylinder, which is the yellow cylinder. Therefore, the total number of objects left in the image is four.

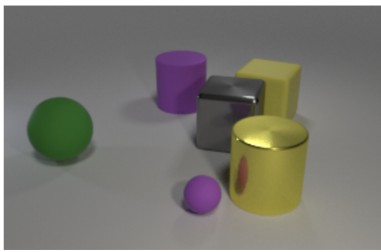

Figure 5: **OneFlow response to visual question.** Implicit visual reasoning in hierarchical generation. OneFlow naturally develops reasoning chains without CoT prompting.

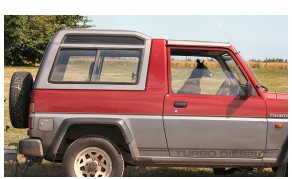

Input image

[CFG 0.0:] A dog sitting in the driver's seat of a truck.

[CFG 1.0:] A dog sitting in the driver's seat of a red and gray truck.

[CFG 2.0:] A red and grey suv truck with a dog sitting in the driver's seat. The truck is parked on a grassy field with a tree on the left side and a clear blue sky in the background.

likely locations for the snowboard. Similarly, for the math puzzle in (Figure 5 bottom), the model first identifies objects in the image that match the prompt description—the green sphere and the large shiny cylinder—before arriving at the final answer.

Our results align with findings in Physics of LLMs (Ye et al., 2024) and MetaMorph (Tong et al., 2024b), where the authors suggest that LLMs precompute reasoning graphs before generating tokens. However, our findings demonstrate that the model can perform the same reasoning chain without autoregressive decoding. This suggests that reasoning capabilities can emerge in non-autoregressive architectures and transfer effectively to OneFlow. We show more example VQA generations compared to the AR baseline in Figure 20.

### 3.4 COMPARISON WITH STATE-OF-THE-ART UNIFIED MODELS

We compare OneFlow with other autoregressive and diffusion multimodal models and summarize the results in Table 2. Since these models were trained on different datasets and with different base LLMs, a controlled comparison is difficult. This is why we trained our own autoregressive multimodal model for a fair, apples-to-apples comparison in the previous section.

OneFlow achieves competitive performance on understanding and generation benchmarks, matching the performance of other state-of-the-art models. For instance, models like MMaDA underwent extensive post-training and reasoning training, while OneFlow did not. Similarly, FUDOKI was initialized from a pretrained multimodal model, whereas ours was trained from scratch.

### 3.5 NEW CAPABILITIES INTRODUCED BY ONEFLOW

#### 3.5.1 CLASSIFIER-FREE GUIDANCE IMPROVES TEXT DETAILEDNESS

The use of continuous-time Markov chains allows us to apply classifier-free guidance (CFG) to our model's insertion rates. Specifically, given an unconditional prediction $\lambda(X_t)Q(X_t)$ and a condi-

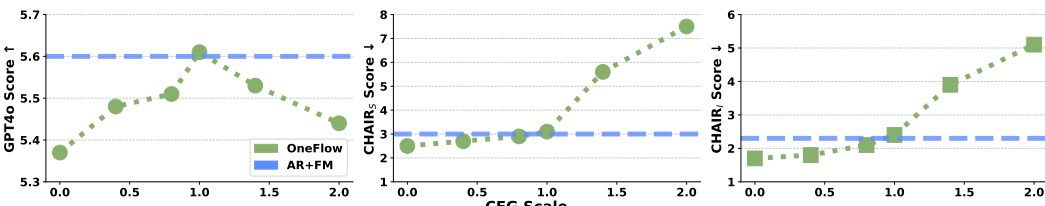

Figure 6: **OneFlow with classifier-free guidance produces longer and more detailed answers** OneFlow matches the AR baseline in GPT-4o as judge (Cheng et al., 2025), while higher guidance scales trade off between detailedness and hallucination. $\text{CHAIR}_S$ and $\text{CHAIR}_I$ measure the rate of hallucinated objects at the sentence and instance levels, respectively (lower is better).

Generated tokens:      $t = 0$      $t = 1$

**I want to paint flowers but I need a photo for inspiration. Can you show me two pictures of flowers and describe them?**

In this first image, a beautiful arrangement of pink flowers is on display, with a total of six flowers spread out in a visually appealing manner.<|image|>The roses are delicate and have a soft, pink hue, which adds to the overall charm of the scene. The bouquet seems to be the main focus of the image, with the flowers captured in a close-up shot, showcasing their intricate details. The display creates an elegant and serene atmosphere, evoking a sense of romance and appreciation for the beauty of nature. This image features a close-up view of a beautiful pink flower sitting in a glass vase on a table.<|image|>The flower appears to be a gerbera, with a vibrant pink color and a prominent center, possibly a bud. The glass vase is positioned at the base of the flower, enhancing the visual appeal of the scene. The table beneath the vase provides a simple yet elegant backdrop, allowing the flower to be the focal point of the image.

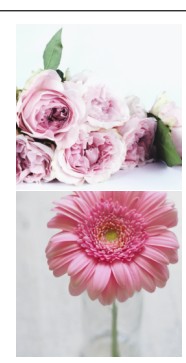

Figure 7: **Concurrent interleaved text & image generation.** OneFlow can insert variable number of images in the generated sequence, which are concurrently denoised alongside the text. This allows the text and images to depend on each other during the generation process.

tional prediction $\lambda(X_t|c)Q(X_t|c)$, where $c$ is the prompt and $w$ is the guidance weight, the modified insertion rate is constructed as:

$$\lambda^{\text{cfg}}(X_t|c) = \lambda(X_t|c)^w \lambda(X_t)^{1-w} \quad \text{and} \quad Q^{\text{cfg}}(X_t|c) \propto Q(X_t|c)^w Q(X_t)^{1-w}. \quad (12)$$

As shown in Figures **??** and 10, higher CFG values consistently increase the length and detail of generated text. We quantitatively evaluated caption quality and hallucination using CapArena (Cheng et al., 2025) by prompting GPT4-o. Our findings show that increasing CFG leads to more detailed captions, with OneFlow matching AR's level of detail at a guidance scale of 1. However, this increased detail comes at the expense of hallucinations at very high CFG values. Figure 6 shows the trade-off between detailness and hallucination across a range of CFG values.

### 3.5.2 SIMULTANEOUS GENERATION OF INTERLEAVED TEXT AND IMAGES

When autoregressive multi-modal models insert an image, they append it at the end of the current generation, fully denoise it, then continue the generation process. However, OneFlow is able to simultaneously denoise images and the text. When the model deems it necessary, it is able to insert an new image in the existing text and denoise it along with the text, as proposed in Section 2.3.

To train this model, we took OneFlow 1B Mixed and finetuned it on the interleaved subset of the Chameleon dataset (Team, 2024) for 20000 steps. This subset contains 17000 examples that interleave both text and image data. Figure 7 shows the generation order of the tokens where two images were generated as part of the answer, with more detailed examples in Appendix A and animated versions in the supplementary material.

## 4 RELATED WORK

**Native Multimodal Models.** Current approaches for unified multimodal models fall into three main paradigms: fully autoregressive (Team, 2024; Wang et al., 2024; Wu et al., 2025), hybrid (Zhou

et al., 2025; Deng et al., 2025; Xie et al., 2024; Ma et al., 2025), and fully diffusion-based (Yang et al., 2025; Swerdlow et al., 2025; Li et al., 2025; Wang et al., 2025). While these models are limited by a fixed generation order or fixed-length output, our approach fundamentally differs by being able to simultaneously generate interleaved content and a variable number of images. For a more comprehensive analysis, see Appendix C.

**Discrete Diffusion and Discrete Flow Matching.** Iterative refinement models, including diffusion (Sohl-Dickstein et al., 2015; Ho et al., 2020) and flow models (Liu et al., 2022; Albergo et al., 2023; Lipman et al., 2024), have been adapted for discrete token spaces. Discrete diffusion models typically learn to reverse a corruption process (Austin et al., 2021; Lou et al., 2024), while discrete flow models transport between two distributions with an interpolating scheme (Campbell et al., 2024b; Gat et al., 2024). Although these frameworks offer a large design space (Shaul et al., 2024; Wang et al., 2025), recent works have predominantly focused on a simplified mask construction (Sahoo et al., 2024; Shi et al., 2024a; Ou et al., 2024; Zheng et al., 2024). This masking framework, however, cannot be easily applied to variable-length and especially simultaneous interleaved generation.

**Edit-based Non-autoregressive Language Models.** Early non-autoregressive models for variable-length generation (Gu et al., 2019a;b; Stern et al., 2019; Reid et al., 2022) often relied on multiple models and evaluations to handle edit operations. While later work like Edit Flows (Havasi et al., 2025) improved on this by using a continuous-time framework and using only a single evaluation per step. Campbell et al. (2024a) also proposed modeling insertions with a diffusion model for denoising, but did not consider sequential data. In contrast, our approach considers sequential mixed-modal data, allows for parallel token insertions, and uses a unified backbone architecture.

## 5   CONCLUSION AND LIMITATIONS

We introduced OneFlow, a novel non-autoregressive multimodal model that overcomes the fixed-length generation limitations of diffusion models and has better scaling than autoregressive multimodal models. We introduced mixed-modal generation approaches, which through extensive controlled experiments, improve on benchmarks for both image understanding and image generation. We also propose a novel approach to interleaved generation that simultaneously denoises images and inserts text tokens, with promising qualitative results. Interleaved generation is still in its infancy and we expect to see more incoming research efforts in constructing large-scale data sets (Awadalla et al., 2024; Laurençon et al., 2023; Zhu et al., 2023) and designing comprehensive benchmarks.

A limitation of requiring bidirectional attention is the lack of key-value caching, which increases inference cost. Although we do find that OneFlow can obtain good captioning performance with very few model evaluations—outperforming AR with only 6 sampling steps (Figure 15)—it is still slower and more memory-intensive compared to key-value cached autoregressive sampling. Reducing inference costs, with semi-autoregressive models (Arriola et al., 2025; Gat et al., 2025) or more sophisticated methods, would be an exciting research direction.

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

# A  ADDITIONAL GENERATION EXAMPLES

| Generated Text | Generated Images |
| --- | --- |

**PROMPT:** I want to paint flowers but I need a photo for inspiration. Can you show me two pictures of flowers and describe them?

**t = 0.2**: first a pink of out in appealing roses and overall the. seems be the, appreciation for.

a pink a flower pink with possibly vase is base the flower appeal simple backdrop the point of the image

No image

**t = 0.4**: In first, a arrangement pink flowers is, of six flowers out in appealing manner.<|image|> roses are delicate and pink the overall of the. The bouquet seems be the captured,oking and appreciation for the nature.
This image a view pink sitting a table flower be ger with a pink with prominent center possibly a. vase is base the flower visual appeal table a simple backdrop, allowing flower be the point of the image.

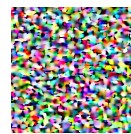

**t = 0.6**: In first, a arrangement of pink flowers is, of six flowers out in a appealing manner.<|image|> roses are delicate and have a, pink, the overall of the. The bouquet seems be focus the image captured, showcasing their The display and atmosphere,oking of and appreciation for the nature.
This image a view of a pink sitting a table.<|image|> flower appears be ger with a pink with prominent center possibly a bud. vase is the base the flower, enhancing visual appeal of the. The table vase a simple backdrop, allowing flower be the point of the image.

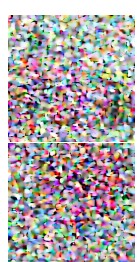

**t = 0.8**: In this first image, a beautiful arrangement of pink flowers is on display, with total of six flowers spread out in a appealing manner.<|image|>The roses are delicate and have a soft, pink hue, which adds the overall charm of the scene. The bouquet seems be the main focus of the image flowers captured a close shot, showcasing their intricate details. The display and atmosphere, evoking a sense of romance and appreciation for the beauty nature.
This image a view of a pink flower sitting a glass a table.<|image|>The flower appears be ger with a pink color with a a prominent center, possibly a bud. vase is the base of the flower, enhancing visual appeal of the. The table the vase provides a simple backdrop, allowing flower to be the focal point of the image.

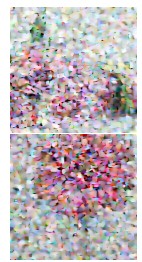

**t = 1.0**: In this first image, a beautiful arrangement of pink flowers is on display, with a total of six flowers spread out in a visually appealing manner.<|image|>The roses are delicate and have a soft, pink hue, which adds to the overall charm of the scene. The bouquet seems to be the main focus of the image, with the flowers captured in a close-up shot, showcasing their intricate details. The display creates an elegant and serene atmosphere, evoking a sense of romance and appreciation for the beauty of nature.
This image features a close-up view of a beautiful pink flower sitting in a glass vase on a table.<|image|>The flower appears to be a gerbera, with a vibrant pink color, with a vibrant pink color and a prominent center, possibly a bud. The glass vase is positioned at the base of the flower, enhancing the visual appeal of the scene. The table beneath the vase provides a simple yet elegant backdrop, allowing the flower to be the focal point of the image.

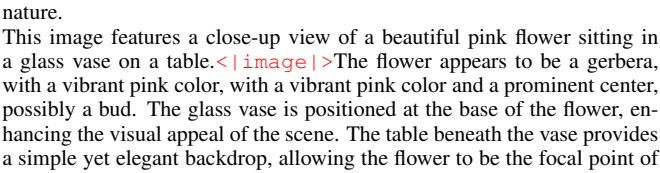
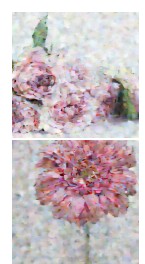

**t = 2.0**: In this first image, a beautiful arrangement of pink flowers is on display, with a total of six flowers spread out in a visually appealing manner.<|image|>The roses are delicate and have a soft, pink hue, which adds to the overall charm of the scene. The bouquet seems to be the main focus of the image, with the flowers captured in a close-up shot, showcasing their intricate details. The display creates an elegant and serene atmosphere, evoking a sense of romance and appreciation for the beauty of nature.
This image features a close-up view of a beautiful pink flower sitting in a glass vase on a table.<|image|>The flower appears to be a gerbera, with a vibrant pink color, with a vibrant pink color and a prominent center, possibly a bud. The glass vase is positioned at the base of the flower, enhancing the visual appeal of the scene. The table beneath the vase provides a simple yet elegant backdrop, allowing the flower to be the focal point of the image.

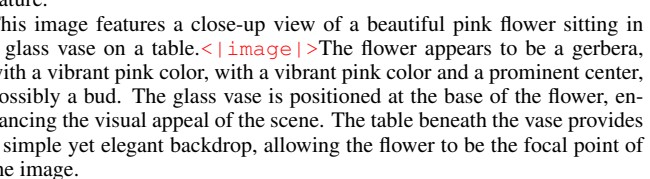
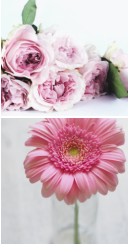

| Generated Text | Generated Images |
|---|---|
| **PROMPT:** I'm looking for a new hobby that's relaxing and will allow me to be outdoors to get some fresh air. Offer me a suggestion and show me an image of the type of place I can do it | |
| $t = 0.2$: ? It's that. senseness tranqu promotes. | No image |
| $t = 0.4$: Why try? It's a that you a wide of, lake trail. in can create senseness tranqu promotes stress and. | No image |
| $t = 0.6$: Why not try nature? It's a that you a wide variety of natural, like this lake trail,. in can create sense ofness and tranqu promotes stress and. | No image |
| $t = 0.8$: Why not try nature? It's a relaxing that allows you to in a wide variety of natural environments, like this lake trail, for instance.`<|image|>`Walking in can create sense of calmness and tranqu that promotes stress and relaxation. |  |
| $t = 1.0$: Why not try nature walking? It's a relaxing hobby that allows you to immerse yourself in a wide variety of natural environments, like this lake trail, for instance.`<|image|>`Walking in nature can create a sense of calmness and tranquility that promotes stress reduction and relaxation. |  |
| $t = 2.0$: Why not try nature walking? It's a relaxing hobby that allows you to immerse yourself in a wide variety of natural environments, like this lake trail, for instance.`<|image|>`Walking in nature can create a sense of calmness and tranquility that promotes stress reduction and relaxation. |  |

Figure 8: Example interleaved generation process showing outdoor exercise. In this example, the image is inserted late in the generation process when the text is almost fully denoised.

| Generated Text | Generated Images |
|---|---|

**PROMPT:** I am teaching an ESL lesson on home types. Please show me images of a bungalow, a two-story home and a townhouse that I can show my students, and briefly explain about each of them.

**t = 0.2**: good. **B is typically for with a-p simple including suburban towns. is of a.Two** type residential building levels with the typically bedrooms on the. example-story. **Town town unit town. They vary typically than homes image example.<|image|>

**t = 0.4**: which good show to them about different types1. **Bungalow** ungalow is of is typically cosy designed for. Bungal are single-story, with a-p simple can including suburban neighbourhood towns and areas. Below is of a.Two-** A a type residential building levels, with the typically on bedrooms on the. below example of-story. **Town A town multi residential unit that town walls. They can vary size are typically smaller than single homes image is example of.<|image|>

**t = 0.6**: images which would good to show students to them about different types:
1. **Bungalow** ungalow is a type of is typically small, cosy and designed for comfortable. Bungal are often single-story, with a-p simple can found various locations including suburban neighbourhood towns and areas. Below is an image of a.Two-** A two a type residential building has levels, with the typically on the first floor and bedrooms and additional on the. image below an example of-story. **Town A town multi-level residential unit that attached town shared walls. They can vary size, are typically smaller than single homes. The image is example of of.<|image|>

**t = 0.8**: are images which would be good to show your ESL students to them about different types:
1. **Bungalow** A bungalow is a type of that is typically small, cosy and designed for comfortable living. Bungalows are often single-story, with a-pitched and a simple floor can be found various locations, including suburban neighbourhoods, beach towns and areas. Below is an image of a b2.Two-Story home** A two home a type residential building has levels or floors, with the typically on the first floor and bedrooms and additional spaces on the second. image below is an example of a two-story home.3. **Townhouse** A town a multi-level residential unit that attached one more other town shared walls. They can vary size, but are typically smaller than single-family homes. The image below is example of a of-houses.<|image|>

**t = 1.0**: Here are three images which would be good to show your ESL students to teach them about different home types:
1. **Bungalow** A bungalow is a type of house that is typically small, cosy, and designed for comfortable living. Bungalows are often single-story homes, with a low-pitched roof and a simple, open floor plan. They can be found in various locations, including suburban neighbourhoods, beach towns, and rural areas. Below is an image of a bungalow:<|image|>2. **Two-Story home** A two-story home is a type of residential building that has two levels or floors, with the main living areas typically on the first floor and bedrooms and additional living spaces on the second floor. The image below is an example of a two-story home.<|image|>3. **Townhouse** A townhouse is a multi-level residential unit that is attached to one or more other townhouses via shared walls. They can vary in size, but are typically smaller than single-family homes. The image below is an example of a row of townhouses.<|image|>

**t = 2.0**: Here are three images which would be good to show your ESL students to teach them about different home types:
1. **Bungalow** A bungalow is a type of house that is typically small, cosy, and designed for comfortable living. Bungalows are often single-story homes, with a low-pitched roof and a simple, open floor plan. They can be found in various locations, including suburban neighbourhoods, beach towns, and rural areas. Below is an image of a bungalow:<|image|>2. **Two-Story home** A two-story home is a type of residential building that has two levels or floors, with the main living areas typically on the first floor and bedrooms and additional living spaces on the second floor. The image below is an example of a two-story home.<|image|>3. **Townhouse** A townhouse is a multi-level residential unit that is attached to one or more other townhouses via shared walls. They can vary in size, but are typically smaller than single-family homes. The image below is an example of a row of townhouses.<|image|>

Figure 9: Example interleaved generation process showing home types. An animated version is included in the supplementary material.

| Input image | Captions with varying classifier-free guidance weights |
|---|---|
|  | **[CFG 0.0:]** A llama and a horse standing in a field. 

**[CFG 2.0:]** A white llama and a white horse standing in a field. |
|  | **[CFG 0.0:]** A group of toy animals sitting on a table. 
**[CFG 1.0:]** A table topped with a variety of toy animals, including a giraffe , a cow , and a bird , as well as a toy duck . In the background, there is a frame attached to the wall. 
**[CFG 1.5:]** A green table topped with toy animals, including a giraffe, a cow, a yellow bird, and a toy duck, next to a red box . In the background, there is a photo frame attached to the wall. |
|  | **[CFG 0.0:]** A bathroom with a large tub and a sink. 
**[CFG 1.0:]** A bathroom with a claw foot tub and three windows . 
**[CFG 2.5:]** A brown bathroom or master bathroom with a classic claw foot tub and three windows. |
|  | **[CFG 0.0:]** A cat sitting on a wooden deck looking up. 
**[CFG 1.0:]** A cat sitting on a wooden deck looking at its reflection in a window. 
**[CFG 2.5:]** Two fluffy ginger and white cats sit and gaze at their reflection in a glass window on a green wooden deck in Japan . |
|  | **[CFG 0.0:]** A white plate topped with a cake and a spoon. 
**[CFG 1.0:]** A plate with a dessert and two spoons on it. 
**[CFG 2.0:]** A white plate topped with ice cream , accompanied by two spoons, a bottle , a glass , and a tissue paper on the table . Through the glass window in the background, we can see the water and the sky. |
|  | **[CFG 0.0:]** A glass bowl filled with colorful paper cranes. 
**[CFG 1.0:]** Colorful origami cranes in a glass bowl shaped like a heart . 
**[CFG 2.0:]** A table with a heart-shaped bowl filled with colorful origami cranes in various colors . The background is slightly blurred , giving the focus to the vibrant colors of the cranes. |

Figure 10: Text generation examples from OneFlow, which allows the use of classifier-free guidance (CFG). We observe that CFG produces longer and more detailed captions and also increased chance of hallucinations . Highlighted text show increased levels of detail when using higher CFG weights.

## B FULL DERIVATIONS

We provide the derivations of the model here. We briefly summarize the Edit Flow (Havasi et al., 2025) formulation and derivation, and then derive the interleaved time schedule when insertions and image denoising are performed simultaneously.

**Setup.** We make use of a blank token $\varepsilon$ to denote empty spaces within a sequence. This token is only used for tracking token deletions during training and is not part of the vocabulary. Let $\mathcal{Z} = \bigcup_{n=0}^{N}(\mathcal{T} \cup \{\varepsilon\})^n$ be an extended space of aligned sequences. Furthermore, define $f_{\text{rm-blanks}} : \mathcal{Z} \to \mathcal{X}$ as the function that removes all blank tokens from the sequence. Lastly, we define the delta function over sequences $\delta_{z_1}(z_2) = \prod_i \delta_{z_1^i}(z_2^i)$ which is one if all tokens are the same otherwise zero (i.e. Kronecker's delta function).

**Continuous-time Markov chain (CTMC).** A CTMC is a continuous-time discrete-space process which iteratively jumps between discrete values, with transitions

$$\mathbb{P}(X_{t+h}|X_t) = \delta_{X_t}(X_{t+h}) + h u_t(x|X_t) + o(h), \tag{13}$$

where $u_t$ can be interpreted as a first-order characterization of the transition kernel. Since with insertions, the sequence lengths of $X_t$ can change over time. To simplify notation, Havasi et al. (2025) used an augmented space of $(X_t, Z_t)$, where it is basically always enforced that $X_t = f_{\text{rm-blanks}}(Z_t)$. The role of $Z_t$ is only for training, to keep track of which tokens are deleted and to compute the loss, and it is neither seen by the model nor used during sampling.

To briefly summarize the construction below, the Flow Matching recipe makes use of a prescribed conditional CTMC that generates single data sequences, which is then marginalized over the data distribution. The resulting marginal CTMC will then sample from the data distribution.

**Conditional probability path.** Given a data sequence $X_1 \sim p_{\text{data}}$, we prescribe a conditional probability path over $Z_t$ of the same sequence length which interpolates between the empty sequence and this data sequence. We then obtain $X_t$ by applying the $f_{\text{rm-blanks}}$ function. Concretely, we can express the conditional probability path as

$$p_t(X_t, Z_t|X_1) = p_t(X_t|Z_t, X_1) \cdot p_t(Z_t|X_1) \tag{14}$$

$$= p_t(X_t|Z_t) \cdot p_t(Z_t|Z_1) \tag{15}$$

$$= \delta_{f_{\text{rm-blanks}}(Z_t)}(X_t) \cdot \left(\prod_{i=1}^{n}(1-\kappa_t)\delta_\varepsilon(Z_t^i) + \kappa_t \delta_{X_1^i}(Z_t^i)\right), \tag{16}$$

where $\kappa_t$ is a scheduler where $\kappa_0 = 0, \kappa_1 = 1$, and $n$ is the sequence length of $X_1$. In English, (16) is a mixture distribution where each token $Z_t^i$ can either be equal to $\varepsilon$ with probability $1 - \kappa_t$ or equal to data value $X_1^i$ with probability $\kappa_t$.

**Conditional CTMC rate.** As discussed in Havasi et al. (2025), a conditional CTMC that samples from this conditional probability path can be constructed as

$$u_t(x, z|X_t, Z_t, X_1) = \left(\sum_{i=1}^{n} \frac{\dot{\kappa}_t}{1 - \kappa_t}(\delta_{X_1^i}(z^i) - \delta_{Z_t^i}(z^i))\right) \delta_{f_{\text{rm-blanks}}(z)}(x), \tag{17}$$

$$\text{where } x = \text{ins}(X_t, i, a) \text{ for some } i \in [n] \text{ and } a \in [M]$$

which denotes the infinitesimal change in probability of going from the state $(X_t, Z_t) \to (x, z)$, constrained to next sequences $x$ that are one token insertion difference from $X_t$. In English, (17) assigns a rate of $\frac{\dot{\kappa}_t}{1-\kappa_t}$ if $Z_t^i$ is not yet equal to $X_1^i$; otherwise, it is zero. This ensures that a sample starting with all blanks $Z_0 = [\varepsilon, ...\varepsilon]$ at $t = 0$ will eventually turn into $X_1$ at $t = 1$. This ratio $\frac{\dot{\kappa}_t}{1-\kappa_t}$ is the infinitesimal rate that each token changes its value, matching the distribution imposed by the scheduler $\kappa_t$, and conditioned on that it is still the $\varepsilon$ token at time $t$.

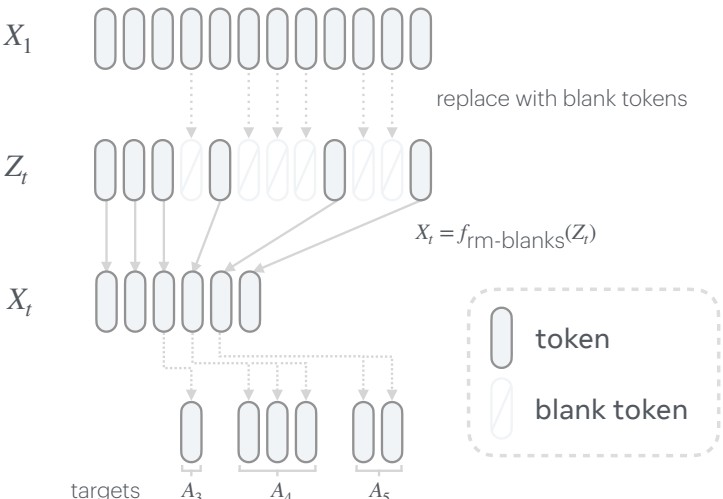

Figure 11: During training we construct $Z_t$ by replacing tokens with the blank token ($\varepsilon$), with the original tokens used to construct the target bag-of-tokens $\mathcal{A}_i$.

**Training loss.** In order to train a model thats transport sequences via insertions,

$$u_t^\theta(x|X_t), \quad \text{where } x = \text{ins}(X_t, i, a) \text{ for some } i \text{ and } a \tag{18}$$

we would need to marginalize out the auxiliary process $Z_t$ and the data $X_1$. Havasi et al. (2025) showed this can be done by using a loss based on any Bregman divergence while summing up over all possible sequences $z$ such that $x = f_{\text{rm-blanks}}(z)$. Concretely, given a convex function $\phi$ that defines a Bregman divergence $D_\phi(a, b) = \phi(a) - \phi(b) - \langle a - b, \frac{\mathrm{d}}{\mathrm{d}b}\phi(b)\rangle$, we can use the loss

$$\mathbb{E}_{X_t, Z_t \sim p_t(X_t, Z_t|X_1), X_1 \sim p_{\text{data}}} D_\phi\Big(\sum_z u_t(\cdot, z|X_t, Z_t, X_1), u_t^\theta(\cdot|X_t)\Big). \tag{19}$$

Plugging in the entropy $\phi(u) = \langle u, \log u\rangle$, this results in the Edit Flow loss

$$\mathbb{E}_{t, p_t(X_t, Z_t|X_1), X_1 \sim p_{\text{data}}} \left[\sum_{x \neq X_t} u_t^\theta(x|X_t) - \sum_{i=1}^{n} \mathbf{1}_{[Z_t^i = \varepsilon]} \frac{\dot{\kappa}_t}{1 - \kappa_t} \log u_t^\theta(\text{ins}(X_t, j, X_1^i)|X_t)\right], \tag{20}$$

where $j$ is the position in $X_t$ that corresponds to the first non-$\varepsilon$ token on the left of $Z_t^i$. This ensures that inserting at the $i$-th position corresponds to changing the value of $Z_t^i$ from $\varepsilon$ to $X_1^i$.

**Loss simplification.** We deviate from Havasi et al. (2025) and use a $t$-independent parameterization. In particular, for sequences $x$ that are one token insertion of $X_t$, i.e., $x = \text{ins}(X_t, i, a)$, we use

$$u_t^\theta\big(\text{ins}(X_t, i, a)|X_t\big) = \frac{\dot{\kappa}_t}{1 - \kappa_t}\lambda^i(X_t)Q^i(a|X_t), \tag{21}$$

where the neural network parameterizes $\lambda$ and $Q$. Using this parameterization, letting $\mathcal{A}_j$ be the set of missing tokens to the right of position $j$ of $X_t$, the training loss (20) can be decomposed into

$$\mathbb{E}_{(\dots)}\left(\frac{\dot{\kappa}_t}{1 - \kappa_t}\right)\left(\sum_{j=1}^{n(X_t)} \lambda^j(X_t) - \sum_{j=1}^{n(X_t)} \sum_{a \in \mathcal{A}_j} \log\big(\lambda^j(X_t)Q^j(a|X_t)\big)\right) \tag{22}$$

$$= \mathbb{E}_{(\dots)}\left(\frac{\dot{\kappa}_t}{1 - \kappa_t}\right)\sum_{j=1}^{n(X_t)}\Bigg(\underbrace{\lambda^j(X_t) - |\mathcal{A}_i|\log\lambda^j(X_t)}_{(4)} + \underbrace{\sum_{a \in \mathcal{A}_j}\log Q^j(a|X_t)}_{(6)}\Bigg) + \text{const.} \tag{23}$$

which recovers the losses for $\lambda$ and $Q$ in (4) and (6) respectively, after removing the coefficient $\frac{\dot{\kappa}_t}{1 - \kappa_t}$. While keeping this coefficient relates the loss the to an evidence lower bound (Havasi et al., 2025), we found that removing this coefficient in the loss gave better results in practice.

## B.1 INTERLEAVED TIME SCHEDULE

In order to model image insertions, we would make a choice. We can either (i) fully denoise images at the time of insertion, or (ii) insert only noise and denoise later. We choose the latter approach, as this allows simultaneous generation across images and text, and provides the best parallelism as only a single model forward at each step is needed for both modalities. Without loss of generality, assume there is only a single image.

Generation starts by advancing the sequence time, denoted $t_{\text{text}} = 0$. When the image is inserted, we associate the image with its own time $t_{\text{img}}$.

The main difficulty is that we can not simply set $t_{\text{img}}$ and $t_{\text{text}}$ independently during training, as evidently we always have $t_{\text{text}} \geq t_{\text{img}}$. In fact, an independent scheduler induces the wrong distribution for our insertion prediction, and it will not insert the correct distribution at generation time. Instead, we need to ensure that training and generation see the same distribution of time values. To achieve this, we first note that the image exists in the sequence according to the scheduler $\kappa_t$, which means that the *insertion times* are distributed according to

$$p(t_{\text{insert}}) = \dot{\kappa}_t, \tag{24}$$

where $t_{\text{insert}}$ is the time at which an image is inserted, *i.e.*, $\kappa_t$ is the cumulative distribution function (CDF) of the insertion times. Equivalently, to sample the insertion time, we can apply the inverse CDF sampling,

$$t_{\text{insert}} = \kappa^{-1}(u), \qquad u \sim \text{Unif}(0, 1). \tag{25}$$

If we set $t_{\text{img}} = 0$ when an image is inserted, then the difference between $t_{\text{text}}$ and $t_{\text{img}}$ is distributed according to the insertion time. This gives us the relation

$$t_{\text{text}} - t_{\text{img}} = t_{\text{insert}} \tag{26}$$

when $0 \leq t_{\text{text}}, t_{\text{img}}, t_{\text{insert}} \leq 1$. Since $t_{\text{text}}$ will reach 1 before $t_{\text{img}}$, and we want to train for the entire process until $t_{\text{img}} = 1$, we can construct an extended time interval

$$\tau_{\text{text}} \in [0, 2], \qquad t_{\text{text}} = \text{clip}(\tau_{\text{text}}), \tag{27}$$

where $\text{clip}(\tau) = \min\{1, \max\{0, \tau\}\}$ clips the time values back into the interval $[0, 1]$.

During training, we first sample $\tau_{\text{text}}$, then sample

$$\tau_{\text{img}} = \tau_{\text{text}} - \kappa^{-1}(u), \qquad u \sim \text{Unif}(0, 1). \tag{28}$$

This will sample an extended time for the image in the internal [-1, 2]. If $\tau_{\text{img}} < 0$, then it has not yet been inserted, hence it is deleted from the sequence. Otherwise, it is clip,

$$t_{\text{img}} = \text{clip}(\tau_{\text{img}}), \tag{29}$$

and we proceed to use the Flow Matching loss (9) to train the image denoising.

| Generative Model Framework | | | | |
|---|---|---|---|---|
| Text | AR | AR | Mask Diffusion | Discrete FM | Edit Flow |
| Image | AR | Diffusion / FM | Mask Diffusion | Discrete FM | FM |
| *Training Properties* | | | | | |
| Attention | Causal | Block Causal | Bidirectional | Bidirectional | Bidirectional |
| Tokens / Iter | Seq Len | up to 2x Seq Len | Seq Len | Seq Len | ~50% Seq Len |
| *Generation Capabilities* | | | | | |
| Understanding | ✓ | ✓ | ✓ | ✓ | ✓ |
| Single image generation | ✓ | ✓ | ✓ | ✓ | ✓ |
| Variable length | ✓ | ✓ | × | × | ✓ |
| Concurrent mixed-modal | × | × | ✓ | ✓ | ✓ |
| Interleaved generation | × | ✓ | × | × | ✓ |
| Examples | Chameleon (Team, 2024) JanusPro (Chen et al., 2025) | Transfusion (Zhou et al., 2025) Bagel (Deng et al., 2025) | MMaDA (Yang et al., 2025) | FUDOKI (Wang et al., 2025) | **OneFlow** |

Table 3: High-level comparison between different frameworks for building unified models of text and image generation.

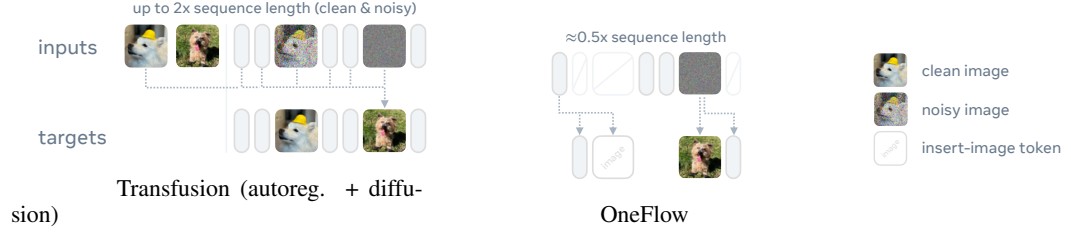

Figure 12: Illustration of the model input and targets during training for (*top*) text generation and (*bottom*) interleaved generation. On text generation, Edit Flow simply deletes tokens instead of replacing them with a special mask token, resulting in lower FLOPS but with the same information as the mask diffusion framework. On interleaved generation, to train autoregressive with diffusion denoising, the images are typically duplicated so that both the clean and the noisy images are in the sequence. On the other hand, OneFlow deletes tokens and images during training which reduces the sequence length.

## C    FRAMEWORK COMPARISON

# D  ONEFLOW ARCHITECTURE

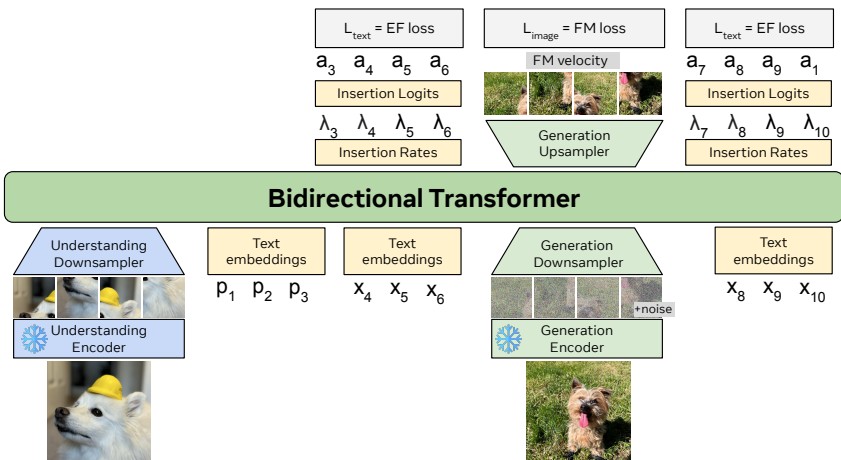

Figure 13: **Architecture.** With a multimodal prompt, OneFlow can produce variable length generations with interleaved text & images in a unified non-autoregressive sequence model, simultaneously generating all modalities with an interleaved time schedule for each generated image and text.

# E ALGORITHMS

---

**Algorithm 1 OneFlow interleaved text–image generation.**

---

1: **function** ONEFLOWGENERATION(step size $\Delta t$, schedule $\kappa$)
2:     $X \leftarrow$ empty sequence                  ▷ *Text tokens (initially empty set)*
3:     $\mathcal{I} \leftarrow \emptyset$                   ▷ *Set of image latents with per-image times*
4:     $t_{\text{text}} \leftarrow 0$
5:     **while** $t_{\text{text}} < 1$ **or** $\exists Y \in \mathcal{I} : t_{\text{img}}(Y) < 1$ **do**
6:         $X, \mathcal{I}, t_{\text{text}}, t_{\text{img}} \leftarrow$ ONEFLOWSTEP$(X, \mathcal{I}, t_{\text{text}}, t_{\text{img}}, \Delta t, \kappa)$
7:     **end while**
8:     **return** $X$ and $\{\text{VAEDec}(Y) : Y \in \mathcal{I}\}$      ▷ *Decode VAE latents into image space*
9: **end function**

---

**Algorithm 2 OneFlow step function.**
$X$ is the token sequence, $\mathcal{I}$ is the set of image latents each with time $t_{\text{img}}(Y)$.

---

1: **function** ONEFLOWSTEP$(X, \mathcal{I}, t_{\text{text}}, t_{\text{img}}, \Delta t, \kappa)$
2:     $(\{\pi, \lambda_{\text{nonzero}}, Q\}, \{v(Y, \cdot)\}_{Y \in \mathcal{I}}) \leftarrow$ OneFlowModel$(X, \mathcal{I}, t_{\text{img}})$

3:     **for all** $Y \in \mathcal{I}$ **with** $t_{\text{img}}(Y) < 1$ **do**      ▷ *Image: Flow matching step on images*
4:         $\Delta t_{\text{img}} \leftarrow \min\{1 - t_{\text{img}}(Y), \Delta t\}$
5:         $Y \leftarrow Y + \Delta t_{\text{img}} \cdot v(Y, t_{\text{img}}(Y))$
6:         $t_{\text{img}}(Y) \leftarrow t_{\text{img}}(Y) + \Delta t_{\text{img}}$
7:     **end for**

8:     $\Delta t_{\text{text}} \leftarrow \min\{1 - t_{\text{text}}, \Delta t\}$
9:     **if** $\Delta t_{\text{text}} > 0$ **then**
10:        **for all** positions $i \in \{1, \ldots, n(X)\}$ **do**      ▷ *Text: parallel insertions*
11:           $p_i^\pi \leftarrow 1 - \pi^i$      ▷ *If using (4) without $\pi$, then skip this step*
12:           $p_i^\lambda \leftarrow \Delta t_{\text{text}} \cdot \frac{\dot{\kappa}(t_{\text{text}})}{1 - \kappa(t_{\text{text}})} \cdot \lambda_{\text{nonzero}}^i$
13:           do-insert $\leftarrow$ Bernoulli$(p_i^\pi)$ and Bernoulli$(p_i^\lambda)$
14:           **if** do-insert **then**
15:              $a \sim Q^i(\cdot \mid X)$
16:              $X \leftarrow \text{ins}(X, i, a)$
17:              **if** $a =$ `<|image|>` **then**
18:                 $Y \sim \mathcal{N}(0, I), \;\; t_{\text{img}}(Y) \leftarrow 0, \;\; \mathcal{I} \leftarrow \mathcal{I} \cup \{Y\}$
19:              **end if**
20:           **end if**
21:        **end for**
22:     **end if**

23:     $t_{\text{text}} \leftarrow t_{\text{text}} + \Delta t_{\text{text}}$
24:     **return** $X, \mathcal{I}, t_{\text{text}}, t_{\text{img}}$
25: **end function**

---

**Algorithm 3** OneFlow training loss with interleaved schedule

1: **function** ONEFLOWTRAININGSTEP(data sequence $X$, image latents $\mathcal{I}$, schedule $\kappa$)
2:      $\tau_{\text{text}} \sim \text{Unif}[0, 2]$
3:      $t_{\text{text}} \leftarrow \min\{1, \tau_{\text{text}}\}$
4:      $j \leftarrow 0$
5:      $X_t \leftarrow []$
6:      **for all** $X^i \in X$ **do**      ▷ *Keep each ground-truth token with prob $\kappa(t_{text})$ to get noisy $X_t$*
7:         **if** $r < \kappa(t_{\text{text}})$ where $r \sim \text{Unif}(0, 1)$ **then**
8:            $X_t \leftarrow X_t + [X^i]$
9:            $j \leftarrow j + 1$
10:            $\mathcal{A}_j \leftarrow \{\}$
11:         **else**
12:            $\mathcal{A}_j \leftarrow \mathcal{A}_j \cup \{X^i\}$      ▷ *Record the deleted tokens at each position in $\mathcal{A}_j$*
13:         **end if**
14:      **end for**
15:      $\mathcal{I}_t \leftarrow \{\}$
16:      **for all** images $Y \in \mathcal{I}$ **do**
17:         $Y_1 \leftarrow \text{VAEEnc(img)}$
18:         $u \sim \text{Unif}(0, 1)$
19:         $\tau_{\text{img}}(Y) \leftarrow \tau_{\text{text}} - \kappa^{-1}(u)$
20:         **if** $\tau_{\text{img}} < 0$ **then**
21:            insert <|image|>in the appropriate $\mathcal{A}_i$      ▷ *Image is "deleted" at this snapshot*
22:         **else**
23:            $t_{\text{img}}(Y) \leftarrow \min\{1, \tau_{\text{img}}(Y)\}$
24:            $Y_0 \sim \mathcal{N}(0, I)$
25:            $Y_t \leftarrow t_{\text{img}}(Y)Y_1 + (1 - t_{\text{img}}(Y))Y_0$
26:            $\mathcal{I}_t \leftarrow \mathcal{I}_t \cup \{Y_t\}$
27:         **end if**
28:      **end for**
                                                  ▷ *Forward pass*
29:      $\{\pi, \lambda_{\text{nz}}, Q\} \leftarrow \texttt{OneFlowModel}(X_t, \mathcal{I}_t)$
                                         ▷ *Compute OneFlow losses*
30:      $\mathcal{L}_{\text{tokens}} \leftarrow \dfrac{1}{n(X_t)} \sum_i \left[ -\sum_{a \in \mathcal{A}_i} \log Q^i(a \mid X_t) \right]$      ▷ *$n(X_t)$ is the length of the sequence*
31:      $\mathcal{L}_{\text{count}} \leftarrow \dfrac{1}{n(X_t)} \sum_i \left( \lambda^i(X_t) - |\mathcal{A}_i| \log \lambda^i(X_t) \right)$
32:      $\mathcal{L}_{\text{img}} \leftarrow \sum_{Y \in \mathcal{I}_t} \mathbf{1}[\tau_{\text{img}}(Y) \geq 0] \cdot \left\| v(Y_t, t_{\text{img}}(Y)) - (Y_1 - Y_0) \right\|_2^2$
33:      $\mathcal{L} \leftarrow \mathcal{L}_{\text{tokens}} + \mathcal{L}_{\text{count}} + \mathcal{L}_{\text{img}}$
34:      $\Theta \leftarrow \texttt{optimizer\_step}(\nabla\mathcal{L}; \Theta)$      ▷ *Compute gradients and update model*
35: **end function**

# F ADDITIONAL EXPERIMENT RESULTS

## F.1 PERFORMANCE BETWEEN AR AND ONEFLOW DURING PRETRAINING

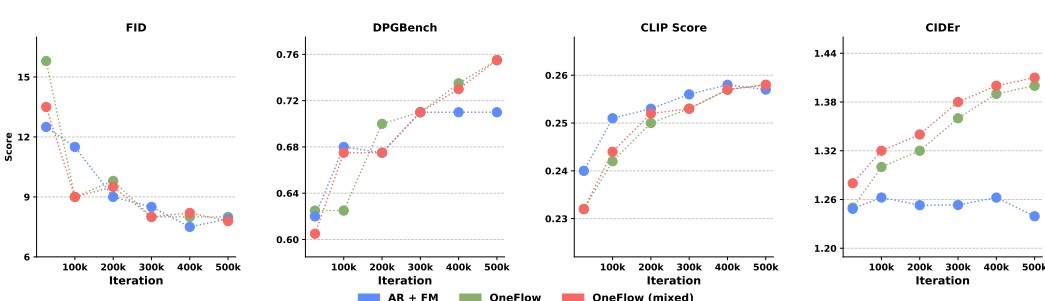

Figure 14: **Training curve for OneFlow vs. AR for multimodal pretraining.** OneFlow initially starts out lower than AR however it quickly catches up and exceeds AR, most notably on DPG and CIDEr.

## F.2 PRETRAINING FROM SCRATCH VS LLAMA INIT

| Model | Initialization | Image Generation | | | CIDEr↑ | VQA |
|---|---|---|---|---|---|---|
| | | DPG↑ | FID↓ | CLIP↑ | | Avg VQA↑ |
| OneFlow | Random | 73.17 | 7.96 | 25.7 | 139.4 | 51.2 |
| OneFlow | LLaMA | 75.41 (+2.24) | 7.79 (−0.17) | 26.0 (+0.3) | 138.2 (−1.2) | 52.2 (+1.0) |
| OneFlow Mixed | Random | 74.86 | 7.69 | 25.8 | 140.0 | 51.6 |
| OneFlow Mixed | LLaMA | 75.08 (+0.22) | 7.44 (−0.25) | 25.8 (+0.0) | 139.1 (−0.9) | 52.8 (+1.2) |
| AR + FM | Random | 71.90 | 7.83 | 25.8 | 122.9 | 46.6 |
| AR + FM | LLaMA | 73.40 (+1.50) | 7.91 (−0.08) | 25.7 (−0.1) | 123.9 (+1.0) | 49.0 (+2.4) |

Table 4: **Ablation study comparing LLaMA initialization vs. random initialization.** Except for CIDEr, using LLaMA as initialization generally offers benefits, especially for dense prompt image generation (DPG) and for VQA performance. Image generation metrics use CFG=3, and VQA results are averaged across benchmarks.

## F.3 SAMPLING STEPS ON CAPTION PERFORMANCE

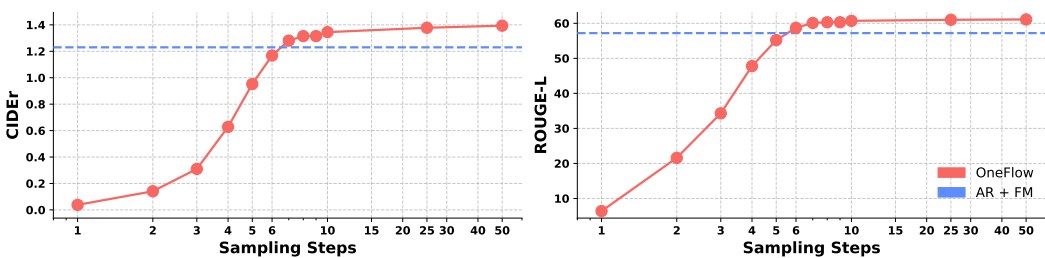

Figure 15: **Performance vs. sampling steps compared to AR.** OneFlow achieves parity with the AR model using only 6 sampling steps.

## F.4 T-INDEPENDENT PARAMETERIZATION

| Time Location | CIDEr |
|:---:|:---:|
| **none** | **1.28** |
| begin_of_seq | 1.27 |
| begin_of_text | 0.33 |

Table 5: CIDEr scores for different time embedding locations.

We empirically found t-independent parameterization to perform better despite theoretical motivation for t-dependence. This likely occurs because $X_t$ already encodes sufficient information about the noise level. We ablated the location of time token insertion: begin_of_seq places it at the beginning of the sequence, begin_of_text at the start of text tokens, and none corresponds to t-independent parameterization.

## F.5 K-SCHEDULER

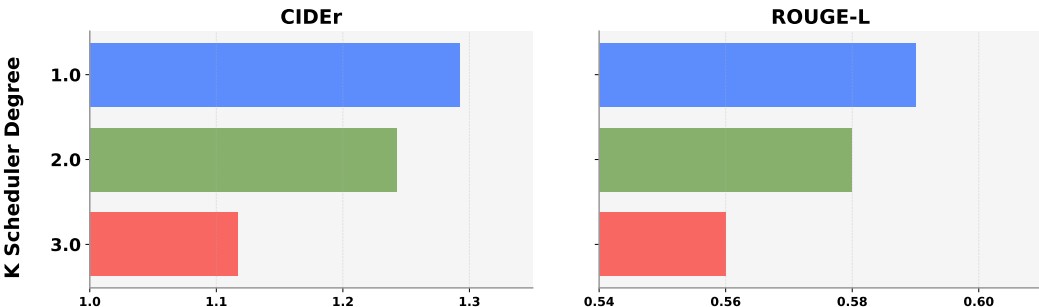

Figure 16: **Effect of $k$-scheduler degree.** Using the schedule $\kappa_t = t^k$, we find that linear scheduling ($k = 1.0$) achieves the best results. Higher degrees lead to overly aggressive token deletion.

We conducted ablations over different k-schedulers. As described in (Gat et al., 2024), there are three variants of the k-scheduler: linear, quadratic, and cubic. We find that the linear scheduler works best.

## F.6 MIXED GENERATION PROBABILITY

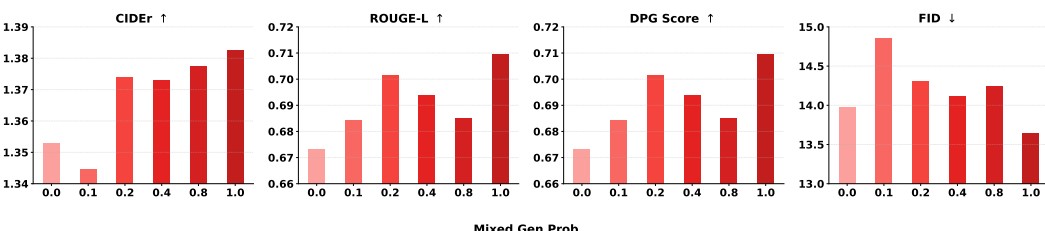

Figure 17: **Effect of mixed-generation probability on performance.** Higher mixed-generation probabilities consistently improve image understanding (CIDEr, ROUGE-L) and compositional generation (DPG Score), while image generation quality (FID) remains stable. This demonstrates that mixed training enhances understanding and compositional capabilities without compromising generation performance.

We ablated different mixed-generation probabilities at 200k training steps. Results show a clear monotonic trend: higher mixed-generation probability consistently improves image understanding across all metrics. CIDEr scores increase from 135.3 (0% mixed) to 138.3 (100% mixed), ROUGE-L improves from 0.67 to 0.71, and DPG Score rises from 0.67 to 0.71. Importantly, image generation quality (FID, DPG Score) remains stable across all probabilities, demonstrating that mixed training strengthens understanding capabilities without degrading generation performance.

## F.7 QUALITATIVE COMPARISON ON IMAGE GENERATION

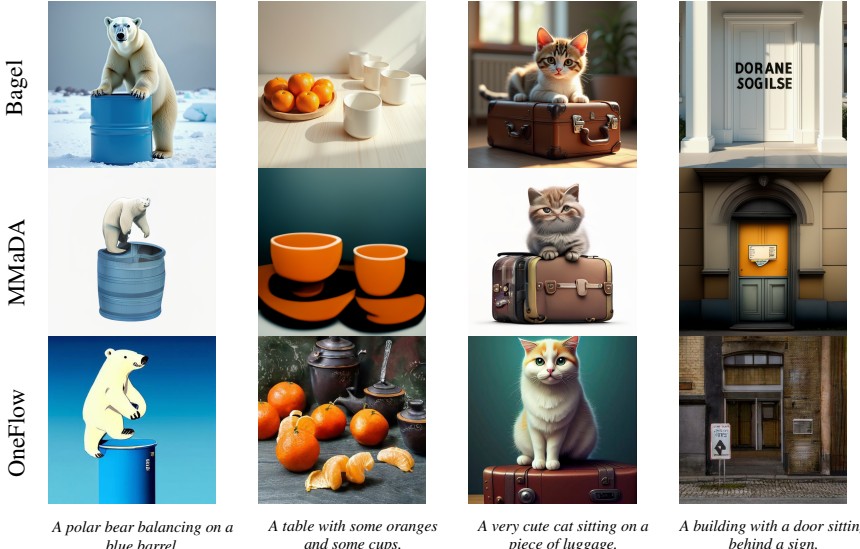

*A polar bear balancing on a blue barrel.*   *A table with some oranges and some cups.*   *A very cute cat sitting on a piece of luggage.*   *A building with a door sitting behind a sign.*

Figure 18: **Qualitative comparison of OneFlow and SOTA models.** We notice that OneFlow gets the details of the prompt correctly, for instance the polar bear is *'balancing on a blue barrel'*. The visual details of our generation are also better compared to MMaDA possibly due to using continuous image tokens rather than discrete. In the last column, the figure shows OneFlow handles common semantic challenges more effectively, as it was able to generate a building with *'a door sitting behind a sign'*.

| Prompt | AR+FM | OneFlow | OneFlow Mixed |
|---|---|---|---|
| *An expansive field, blanketed by the soft light of morning, cradles a collection of eight cabbages, their green heads round and plump. These vegetables are nestled among rows of rich soil, dotted with glistening droplets of dew that cling to their crinkled leaves. As wisps of mist begin to lift, the cabbages lie poised, ready for the day's impending harvest.* | | | |
| *An elegant and modern bathroom featuring a sleek, white rectangular bathtub filled with a froth of soap bubbles. The bathtub rests upon a floor of gray, matte tiles that complement the room's minimalistic design. Against the room's far wall stands a large window that frames the warm, amber hues of a sunset, casting a tranquil glow throughout the space.* | | | |
| *The image captures a whimsical scene with a brown tabby cat, its fur patterned in shades of dark brown, black, and light taupe. The cat, situated as if in the throes of space, is portrayed with a transparent, gleaming bubble encasing its head like an astronaut's helmet. Around it, an assortment of smaller bubbles float serenely in the imagined cosmos, with a creatively interpreted Saturn adorned with rings in the backdrop, providing an aura of interstellar exploration.* | | | |

Figure 19: **OneFlow vs. AR on DPG Bench.** OneFlow outperforms AR significantly on DPG Bench. Through our inspection, we notice that OneFlow gets the subtle details correctly. We highlight the subtle details that the OneFlow model gets correctly in red whereas the AR misses. For the first row, the OneFlow model generates the *'field of cabbages in the soft morning light'*. The second row, OneFlow model generates *'a window with the amber hue'* whereas AR did not. Lastly, the *'brown cat has shades of brown, black and light taupe'* in the OneFlow generations.

## F.8 QUALITATIVE COMPARISON ON IMAGE UNDERSTANDING

### Counting

**Question:** Subtract all large rubber spheres. Subtract all big shiny cylinders. How many objects are left? Explain your answer:

**Ground Truth:** 4

There are five objects left in the image. To find their count, we look at the positions of the spheres relative to the cylinders and the cube. From left to right, the first sphere is the green, the second is the purple, the third is the blue, and the fourth is the brown. The fifth object, which is the yellow cylinder, is not counted since it is not listed in the options. Therefore, the total count of objects that are left with the categories (green, purple, blue, brown) is three.

In the image, there is one object that matches the description of the large sphere: the green sphere, and the only large shiny cylinder, which is the yellow cylinder. Therefore, the total number of objects left in the image is four.

AR                                         OneFlow

(a) **OneFlow compared to AR model on counting task.** OneFlow first locates the object in the prompt before performing the calculation.

### RealWorld Perception

**Question:** In which direction is the front wheel of the car on the right side facing? You must explain your answer. Explain your answer:

**Ground Truth:** Right

The front wheel of the car on the right side is facing towards the camera's perspective. This is determined by the direction of the road, the orientation of the vehicles, and the position of the wheels and headlights. If facing the right side, the direction of the camera is towards the front of the image, then the front wheel of the car is forward, facing the right side of the image. Conversely, if we consider the front of the car to the left side of the image, which is the perspective of the car is facing to the left. Therefore, from the perspective of the camera, and considering the orientation of the front of the car, the front wheel of the car is facing towards the right side of the image.

In the image provided, the front wheel of the car on the right side is aimed towards the left. The wheel is partially visible and appears to be in the process of turning, as indicated by the position of the car and the blurred scenery. Given the context of the street and the position of the car, it is likely that the front wheel is facing towards the direction the car is turning. However, due to the tire's orientation and the tire's angle, it's not possible to determine the steering direction with absolute certainty. The most probable action, based on the visible evidence, is that the front wheel is turned to the left.

AR                                         OneFlow

(b) **OneFlow compared to AR on hard perception task.** AR commits to the answer at the start of the sequence.

### Visual Search

**Question:** In which direction is the front wheel of the car on the right side facing? You must explain your answer. Explain your answer:

**Ground Truth:** Right

The plastic stool in the image is located on the right side of the canal, near the side of the canal. It is blue and stands out against the natural colors of the scene, contrasting with the red colors of the buildings and the canal.

There is no plastic stool visible in the image. The scene includes a canal, boats, trees, and traditional buildings, but none of these items appear to be present within the frame. The stool in question may be part of the rustic outdoor setting or an unrelated object not in the frame.

AR                                         OneFlow

(c) **OneFlow compared to AR on visual search task.** OneFlow first locates the target object (stool), then analyzes its color contrast against the background to determine the answer. Red arrow added for visual clarity (not present in original image).

Figure 20: Comparison of VQA generation between AR and OneFlow.

## G    LLM DISCLOSURE

We used LLMs to create latex code for the figures, generate code, and to improve writing.

---

**Template prompt of VLM-as-a-Judge**

Please act as an impartial judge and evaluate the quality of the following caption provided by an AI assistant for a given image.

Begin your evaluation by providing a short explanation. Be as objective as possible. After providing your explanation, please rate the response on a scale of 1 to 10 by strictly following this format: "[[rating]]", for example: "Rating: [[5]]".

Below are some guidelines for your reference:

1. **Precision**: The caption should accurately correspond to the content of the image, providing precise information about it. Common examples of imprecision include errors in color, quantity, spatial relationships, or the posture of people.

2. **Informativeness**: Salient information in the image should be reflected in the caption. Since it is impossible to include every detail, you will need to subjectively judge which aspects of the image are important. For instance, describing an otter as "a small animal" is precise, but it is less informative than specifying "an otter".

3. **Hallucination**: Captions that include descriptions of objects or elements that are clearly absent from the image should be significantly penalized.

4. **Attention to detail**: Annotators should pay close attention to the details in the image to distinguish the quality of the descriptions.

5. **Assistive description**: Imagine a visually impaired person asking you to describe the image for them. How would you convey the image to them?

6. **Reverse thinking**: What image does the caption lead us to imagine? Does the caption effectively lead you to imagine the intended image?

Image: <image> Reference Caption: <reference caption> Caption: <caption text>

Figure 21: Template prompt of VLM-as-a-Judge, taken from Cheng et al. (2025)