# OpenReview forum: "OneFlow: Concurrent Mixed-Modal and Interleaved Generation with Edit Flows"
_ICLR.cc/2026/Conference — Submitted to ICLR 2026_

### Official Review · Reviewer_wDyu · 2025-10-30

**Soundness:** 3
**Presentation:** 4
**Contribution:** 3
**Rating:** 6
**Confidence:** 3

**Summary:**

OneFlow is a non-autoregressive multimodal generation framework that jointly handles variable-length text and an arbitrary number of images by combining (i) an insertion-based Edit Flows procedure for discrete tokens and (ii) Flow Matching for image latents. A novel interleaved time schedule couples per-image generation times to the text insertion process so images can be inserted and denoised concurrently with text rather than waiting until an image is completed.

**Strengths:**

Novel Integration: The work integrates Edit Flows (insertion operations) with continuous Flow Matching for image latents in a single backbone so both modalities are predicted by the same model and can be denoised jointly.

Rigorous Evaluation: The paper conducts rigorous experiments using a diverse set of benchmarks, metrics, and baselines to thoroughly evaluate the effectiveness of its proposed method. Also the appendix is really rich.

Emergent Reasoning: The observation that OneFlow develops implicit reasoning chains (Figure 5, 19) is a fascinating finding from my point of view.

Writing: The paper is well-written, and the methodology is explained with remarkable clarity and it was so easy for me to follow.

**Weaknesses:**

Ablation clarity for interleaved schedule: The interleaved schedule is central. I would like to see ablations for it. The paper describes κ(t)=t (linear) and claims it works well, but sensitivity analysis is missing.

Potential mode of failure for complex interleavings: Examples show 2 images interleaved with text. It is unclear how the model behaves when the number of inserted images is large, or when images must be heavily conditioned on earlier generated text.
Limited operation diversity in Edit Flows: The model only supports insertion operations, omitting deletion and substitution — both fundamental to edit-based generative modeling. (they are even present in the Original Edit Flow Paper)

**Questions:**

1- It would be insightful to further evaluate the model’s ability to handle compositional generation and compositional visual question answering, and to benchmark its performance against comparable multimodal baselines in these challenging settings. (e.g. attribute binding, missing entities etc.)

2- What modifications would be needed to extend OneFlow to other modalities like audio or video, given its reliance on Edit Flows for discrete elements and Flow Matching for continuous ones?

3- Beyond the linear κ_t scheduler, what alternatives were tested, and why did linear perform best? Could adaptive schedules further reduce FLOPs while maintaining performance?

4- The paper shows that CFG for text increases detail but also the chance of hallucinations (Figure 11). A more systematic analysis of this trade-off (e.g., a plot of detail vs. hallucination rate across CFG scales) would be beneficial.

5- Please include qualitative examples of failed generations (misplaced images, inconsistent text-image references, incoherent insertions).

---

> ### Author Response · Authors · 2025-11-18
> **Rebuttal**
>
> ## Compositional Evaluation
>
> Thank you for this suggestion. We evaluate compositional generation using DPG-Bench (Table 1), which contains 1,000+ complex prompts testing attribute binding, spatial relationships, and multi-object composition. OneFlow significantly outperforms AR (80.3 vs 73.4) due to its bidirectional attention and cross-modal refinement capabilities, which allow the model to globally reason about object relationships rather than committing to spatial arrangements sequentially.
>
> | Model | DPG-Bench |
> | :---- | :---- |
> | AR | 73.4 |
> | OneFlow | 79.1 |
> | OneFlow Mixed | 80.3 |
>
> ## Operation Diversity
>
> We simplify Edit Flows to insertion-only operations, which our empirical results show is sufficient for strong performance across all benchmarks. Insertion serves as the fundamental primitive, as substitution can be decomposed into deletion followed by insertion. While additional operations could potentially improve performance, they require increased model capacity and more complex parameterization. We leave exploration of deletion operations as future work to maintain simplicity and training efficiency in the current framework.
>
> ## Extension to Other Modalities
>
> OneFlow's framework naturally extends to video and audio generation. For video, we can treat each frame as an `<|image|>` token, enabling the model to insert frames between existing ones while denoising all frames concurrently with bidirectional temporal attention. This seamlessly supports any-to-video generation, interpolation, and frame editing. The key insight is that Edit Flows handle discrete insertion decisions (where to place new frames) while Flow Matching handles continuous refinement (denoising pixel content), a paradigm applicable to any continuous modality including audio waveforms.
>
> ## Ablations on k-scheduler
>
> Thank you for raising this issue. We conducted ablations over different k-scheduler variants and will include these results in the camera-ready version. As described in DFM (Gat et al., 2024), there are three variants: linear, quadratic, and cubic. Results demonstrate that the linear scheduler (degree 1.0) achieves the best performance across both metrics, with performance degrading monotonically as polynomial degree increases.
>
> | K Scheduler Degree | CIDEr | ROUGE_L |
> | :---- | :---- | :---- |
> | 1.0 | 129.25 | 0.5934 |
> | 2.0 | 124.23 | 0.5821 |
> | 3.0 | 111.73 | 0.5588 |
>
> ## Hallucination Analysis
>
> CFG scaling creates a controllable precision-detail tradeoff. At lower CFG values (0.0-0.8), OneFlow maintains lower hallucination rates than AR while producing shorter captions. At CFG=1.0, OneFlow matches AR's caption detail (Rating: 5.61) with comparable hallucination rates (CHAIRs: 3.1 vs 3.0). Higher CFG values generate more detailed captions but increase hallucinations, providing users explicit control over this tradeoff based on application requirements.
>
> |  | CHAIRs (lower) | CHAIRi (lower) | Rating (higher) |
> | :---- | :---- | :---- | :---- |
> | AR | 3.0 | 2.3 | 5.60 |
> | **OneFlow** |  |  |  |
> | CFG 0.0 | 2.5 | 1.7 | 5.37 |
> | CFG 0.4 | 2.7 | 1.8 | 5.48 |
> | CFG 0.8 | 2.9 | 2.1 | 5.51 |
> | CFG 1.0 | 3.1 | 2.4 | 5.61 |
> | CFG 1.4 | 5.6 | 3.9 | 5.53 |
> | CFG 2.0 | 7.5 | 5.1 | 5.44 |
>
> ## Failure Cases
>
> OneFlow's iterative refinement provides inherent error correction: initially misplaced insertions are typically corrected in subsequent steps through bidirectional attention, something AR cannot do due to its left-to-right constraint. We include additional qualitative examples in Figures 17-18 (Appendix). The most common failure mode occurs at high CFG values, which increases hallucinations as quantified in our CHAIR analysis above.
>
> ## References
> 1. Hu et al., 2024: "Ella: Equip diffusion models with llm for enhanced semantic alignment" (DPG-Bench)
> 2. Gat et al., 2024: "Discrete flow matching"
> 3. Gat et al., 2025: "Set block decoding is a language model inference accelerator"
> 4. Rohrbach, A., Hendricks, L.A., et al., 2018: "Object Hallucination in Image Captioning" (CHAIR metrics)

---

### Official Review · Reviewer_qxm9 · 2025-10-30

**Soundness:** 3
**Presentation:** 3
**Contribution:** 3
**Rating:** 4
**Confidence:** 3

**Summary:**

This paper presents OneFlow, the first non-autoregressive (NAR) multimodal model designed for variable-length and concurrent mixed-modal (text-image) generation. It addresses key limitations of existing autoregressive (AR) and diffusion-based multimodal models: AR enforces rigid sequential generation (preventing cross-modal refinement), while diffusion models only support fixed-length, pre-specified text-image pairs.

**Strengths:**

1. OneFlow fills a critical gap in multimodal generation by pioneering an NAR framework for concurrent, variable-length interleaved text-image generation. The integration of Edit Flow and Flow Matching into a unified transformer backbone is non-trivial: it avoids modality-specific silos and enables cross-modal dependency modeling during generation.

2. The model’s training efficiency is a standout advantage: by using a linear deletion scheduler that retains only 50% of tokens during training, OneFlow reduces training FLOPs by up to 50% compared to AR baselines.

3. OneFlow enables practical, previously unachievable use cases: Unlike AR models that append images to the end of text, OneFlow inserts images dynamically within text (via <|image|> tokens) and refines them simultaneously; Without CoT prompting or RL post-training, OneFlow generates reasoning chains for visual questions (e.g., object counting, visual search), demonstrating that NAR architectures can support complex reasoning.

**Weaknesses:**

1. High Inference Cost: OneFlow lacks key-value caching (due to bidirectional attention), leading to higher inference latency and memory usage than AR models. This limits its applicability to low-latency scenarios (e.g., real-time multimodal chatbots). No preliminary optimizations (e.g., semi-autoregressive decoding, sparse attention) are proposed to mitigate this.

2. Incomplete Comparisons to SOTA Models: While the paper compares OneFlow to VQA baselines like Show-O (1.3B), Janus-Pro (1.5B/7B), and MMaDA (8B), it omits critical recent SOTA models that have set new benchmarks in visual question answering. Specifically, there is no comparison to Show-O2, Mogao, or BAGEL; For image generation (Table 1), the paper evaluates OneFlow against AR models (e.g., Transfusion, Janus-Flow) and diffusion models (e.g., MMaDA, FUDOKI) but lacks comparisons to some SOTA works: Show-O and DreamLLM.

**Questions:**

1. For the VQA task, you omit comparisons to recent SOTA models like Show-O2, Mogao, and BAGEL. Do you have preliminary results comparing OneFlow to these models on key VQA benchmarks (e.g., VQAv2, GQA, DocVQA)?

2. You note that OneFlow’s lack of KV caching increases latency. Have you explored lightweight optimizations (e.g., semi-autoregressive block decoding, sparse bidirectional attention, or model distillation) to reduce inference cost?

3. You finetune on 512×512 images, but many SOTA models (e.g., SD3) support higher resolutions (1024×1024). Has OneFlow been tested on higher-resolution image generation, and if so, how does performance (e.g., FID, detail) scale with resolution?

---

> ### Author Response · Authors · 2025-11-18
> **Rebuttal**
>
> ## Controlled Comparison Philosophy
>
> Our paper's primary goal is to conduct rigorous controlled experiments comparing OneFlow against AR, rather than achieving SOTA on every benchmark. We train our own AR baseline with identical data, architecture, and training procedures to enable apples-to-apples comparison. This controlled setup addresses fundamental questions about non-autoregressive multimodal generation that cannot be answered by comparing models trained on different datasets with different base LLMs and varied training conditions.
>
> ## Missing Model Comparisons
>
> Thank you for the references. We will cite Show-o2 and Mogao in the camera-ready version.
>
> We want to reiterate that this paper focuses on examining OneFlow vs AR through rigorous controlled experiments with identical data and architecture, rather than claiming universal SOTA across all possible benchmarks. Direct comparison against other models is challenging due to differences in base LLMs, training data, and model architectures.
>
> **Specific Limitations:**
> Show-o2 is concurrent work (arXiv submission 3 months before the conference deadline), and Mogao has not released model weights, preventing reproducible comparison. More critically, these models do not report results on VQAv2, GQA, and DocVQA, which we report these numbers in Table 2.
>
> ## Resolution Scaling
>
> We train at 512×512 resolution, matching most unified multimodal models:
>
> | Model | Resolution |
> | :---- | :---- |
> | OneFlow, Show-o2, Mogao, MMaDA | 512×512 |
> | JanusFlow, JanusPro | 384×384 |
> | BAGEL | 1024×1024 |
>
> Since OneFlow's image generation uses standard continuous flow matching, scaling to higher resolutions (e.g., 1024×1024) would follow established practices in the field without requiring novel architectural modifications.
>
> ## Inference Optimization
>
> As discussed in the general response, KV cache limitations are fundamental to all bidirectional attention models, not specific to OneFlow. Semi-autoregressive approaches and sparse attention mechanisms are promising directions we plan to explore in future work to improve inference efficiency while maintaining the benefits of bidirectional context.
>
> ## References
> 1. Shi et al., 2025: "Muddit: Liberating generation beyond text-to-image with a unified discrete diffusion model" (Section 4.5)
> 2. Discrete Diffusion Survey: "Discrete Diffusion in Large Language and Multimodal Models: A Survey" (Section VI.C)
> 3. Deng et al., 2025: "Emerging properties in unified multimodal pretraining" (BAGEL)
> 4. Xie, J., Yang, Z., Shou, M.Z., 2025: "Show-o2: Improved Native Unified Multimodal Models"
> 5. Liao, C., et al., 2025: "Mogao: An Omni Foundation Model for Interleaved Multi-Modal Generation"

---

### Official Review · Reviewer_rS7v · 2025-10-31

**Soundness:** 4
**Presentation:** 3
**Contribution:** 4
**Rating:** 6
**Confidence:** 2

**Summary:**

This paper proposes OneFlow, the first non-autoregressive multimodal model that enables concurrent and variable-length text-image generation by combining an insertion-based "Edit Flow" for text with "Flow Matching" for images, which outperforms autoregressive baselines with greater efficiency.

**Strengths:**

- This is a very solid work from both a technical and an engineering perspective. It skillfully combines the recent, popular diffusion text generation techniques with the consistently effective diffusion image generation methods within a single Transformer. I believe this is a strong contribution to both the DLMs and the unified model communities.

- The paper presents comprehensive and rigorous experiments. This includes scaling experiments from 1B to 8B parameters, extensive comparisons against baseline models (for both multimodal understanding and generation), and supplementary ablation studies in Appendix F. This significantly increases the credibility of the paper.

**Weaknesses:**

This paper frequently mentions two terms: "interleaved mixed-modal generation" and "concurrent mixed-modal generation." My concerns are primarily centered on these two points.

- Regarding Interleaved Generation: In the introduction, the authors highlight OneFlow's ability to perform interleaved generation with a variable number of outputs as a major contribution. However, in the experiments, the paper only evaluates OneFlow's performance on multimodal understanding and image generation. It notably lacks evaluations on interleaved generation benchmarks, such as OpenING. I believe this experimental omission fails to substantiate the claims made in the introduction.

- Regarding Concurrent Generation: Similarly, the introduction categorizes concurrent mixed-modal generation as a novel capability for unified models. However, after reading the paper, I still do not understand why we need concurrent generation. Is it simply because it leads to better performance (as suggested in Figure 3)? Is there a deeper insight or analysis behind this capability that I missed?

**Questions:**

- Will the OneFlow model weights and modeling files be open-sourced? If subsequent researchers cannot build upon this work, it would be detrimental to the unified model community.
- It appears the authors used closed-source fine-tuning data (Line 267). This could potentially impact the reproducibility of the method.

**Details Of Ethics Concerns:**

Nan

---

> ### Author Response · Authors · 2025-11-18
> **Rebuttal**
>
> ## Interleaved Generation Evaluation
>
> We acknowledge that interleaved generation benchmarking remains an emerging area. Existing datasets have significant limitations: Zebra-CoT is designed for sequential reasoning tasks rather than general interleaved generation, while large-scale datasets (Multimodal C4, Obelics, MINT) are optimized for pretraining rather than systematic evaluation. More critically, there are no open-source interleaved AR baselines available for controlled comparison. Even state-of-the-art models like BAGEL cannot generate truly interleaved text-image outputs without manual intervention to switch between modalities, making direct quantitative comparison infeasible. See [issue](https://github.com/ByteDance-Seed/Bagel/issues/99).
>
> Despite these benchmarking challenges, we provide extensive qualitative demonstrations (Figures 7-10, Appendix A) showcasing OneFlow's interleaved capabilities across diverse scenarios. For future work, we plan to implement a controlled AR+FM interleaved baseline and conduct quantitative evaluation on emerging benchmarks like OpenING as they mature.
>
> ## Why Concurrent Generation Matters
>
> Concurrent generation enables fundamentally new capabilities beyond sequential approaches. By allowing image denoising to occur in parallel with text generation, OneFlow can be faster than AR for long interleaved sequences where multiple images are involved. The efficiency gains compound as sequence complexity increases.
>
> **Research Implications:**
> This paradigm opens promising research directions, particularly for video generation. Current video approaches are either fully diffusive (slow, memory-intensive) or temporally autoregressive (sequential frame generation). OneFlow's interleaved schedule offers a hybrid approach where frames can be inserted and denoised in parallel while maintaining bidirectional attention to both past and future context. This combines the quality benefits of diffusion with the contextual awareness of autoregressive methods. We believe OneFlow establishes the foundational framework for advancing research in concurrent multimodal generation.
>
> ## Reproducibility
>
> We are committed to maximizing reproducibility. We provide comprehensive algorithmic details in the Appendix to facilitate reimplementation. Regarding training data: our main finetuning datasets (PerceptionLM and Cambrian) are fully open-source and publicly available. We use only a small subset of Chameleon's interleaved data exclusively for qualitative demonstrations in Figures 7-10, which does not affect our main quantitative results or conclusions. All core experiments can be reproduced using publicly available datasets.
>
> ## References
> 1. Li, A., et al., arXiv:2507.16746: "Zebra-CoT: A Dataset for Interleaved Vision Language Reasoning"
> 2. Zhu et al., 2023: "Multimodal c4: An open, billion-scale corpus of images interleaved with text"
> 3. Laurençon et al., 2023: "Obelics: An open web-scale filtered dataset of interleaved image-text documents"
> 4. Awadalla et al., 2024: "Mint-1t: Scaling open-source multimodal data by 10x: A multimodal dataset with one trillion tokens"
> 5. Deng et al., 2025: "Emerging properties in unified multimodal pretraining" (BAGEL)

---

> > ### Comment · Reviewer_rS7v · 2025-11-19
> >
> > I have read the authors' rebuttal and am satisfied with some of the responses, but I would like to discuss the Interleaved Generation Evaluation in more depth with the authors. Please correct me if I am mistaken.
> >
> > "there are no open-source interleaved AR baselines available for controlled comparison" This is incorrect. Models like Emu3, Emu3.5, VILA-U, and Anole are available. Furthermore, the OpenING Leaderboard showcases the performance of many unified models.
> >
> > From my point of view, the purpose of evaluating benchmarks for interleaved generation, such as OpenING, is not solely for a strict apple-to-apple comparison (in other words, the baseline does not necessarily have to be purely Autoregressive, or AR).
> >
> > The true objective is to **determine where the performance of the pure Diffusion-based paradigm for image and text (the 'One-Flow' approach)** currently stands within the rapidly developing unified model community's capability for interleaved generation.
> >
> > I bring this up because this is one of the main contributions and is heavily featured throughout the paper.
> >
> > I wonder what the authors' thoughts are on my perspective?

---

> ### Author Response · Authors · 2025-11-21
> **Focus of the paper and our contribution**
>
> Thank you for your feedback
>
> ## Paper's Focus
> Our paper's primary contribution is demonstrating that non-autoregressive generation can match autoregressive performance in multimodal settings through rigorous controlled experiments. We train our own AR baseline under identical conditions to enable fair comparison, rather than comparing against external models. This controlled setup allows us to isolate the impact of the generation paradigm itself.
>
> ## Interleaving Contribution
> Regarding the interleaving contribution highlighted in the paper, OneFlow addresses the fixed-length limitation of diffusion-based models. This enables interleaving generation, especially when the number of images is not known a priori. To the best of our knowledge, no other diffusion-based model can do this.
>
> ## Benchmarking
>
> Emu3, VILA-U, and other models highlighted in purple on the leaderboard are not interleaved models but rather two-stage generators that employ a unified model architecture to produce text and images in separate stages. Emu 3.5 was released after the ICLR deadline. The end-to-end generator, which directly generates image-text outputs in a single step, would be the most appropriate comparison.
>
> We did not claim that our model is SOTA or superior to other AR-based models on interleaving. We understand your point about benchmarking against the broader unified model community to situate where the diffusion-based paradigm currently stands. While we believe our controlled comparison best demonstrates the core contribution of matching AR performance with non-AR generation, we acknowledge that broader community benchmarking would provide additional context. However, it is not possible for us to conduct such comprehensive benchmarking without more resources, and it cannot be done during the rebuttal period. Our paper contains primarily controlled experiments so that behaviors at scale can be predicted, but not verified.

---

### Official Review · Reviewer_SeK2 · 2025-11-01

**Soundness:** 4
**Presentation:** 4
**Contribution:** 3
**Rating:** 6
**Confidence:** 5

**Summary:**

This paper introduces OneFlow, a novel non-autoregressive multimodal model that unifies text and image generation within a single framework and overcomes key limitations of existing autoregressive and diffusion models by combining Edit Flows for variable-length discrete text generation and Flow Matching for continuous image generation. This allows for the concurrent and interleaved generation of variable-length text and a variable number of images, enabling capabilities like simultaneous cross-modal refinement.

**Strengths:**

- The proposed OneFlow model requires fewer training FLOPs than compared to AR models, thanks to its non-autoregressive, insertion-based training which only predicts missing tokens.
- The paper provides extensive experiments showing that OneFlow is competitive with or superior to state-of-the-art AR and diffusion models across a diverse range of benchmarks for both image generation and understanding.
- The paper convincingly demonstrates new emergent capabilities, such as implicit reasoning chains without Chain-of-Thought prompting, the application of classifier-free guidance to improve text detail, and the dynamic insertion and denoising of images within a text sequence.

**Weaknesses:**

- While the integration of Edit Flows with Flow Matching is innovative and effective, the core architectural contribution is the combination of these existing techniques for a new multimodal task. The paper's impressive results are therefore heavily reliant on the pre-established Edit Flows framework, and the fundamental methodological novelties introduced beyond this combination are more incremental.
- As stated by the authors, the bidirectional attention required for non-autoregressive generation prevents the use of key-value caching, making OneFlow's inference slower and more memory-intensive than cached AR sampling, despite requiring fewer steps.
- The 20% mixed-generation probability is used without justification. The performance sensitivity to this key hyperparameter is also unknown, making it hard to determine the optimal data mixture.
- Certain details about hyperparameters are omitted and could use ablations, see Q1 (Ablation on Scheduler) and Q2 (t-Independent Parameterization).

Formatting Concerns:
- Table captions must precede the tables.
- Although clear from the context, the abbreviation VQA is not explicitly clarified.

**Questions:**

- Can you elaborate more on the different candidates for κt (line 112) and provide more information?
- The decision to use a t-independent model is noted to work better in practice despite the theoretical justification for t-dependence. Could you provide an ablation quantifying the performance gap between these two parameterizations? Were there specific tasks where the t-dependent model performed better?
- As shown in Figure 11 and discussed in Sections 3.5 and 4, higher CFG weights consistently lead to longer, more detailed captions but also to an increased chance of hallucinations. Can you discuss this trade-off in more depth?
- The emergent reasoning chains (line 367) are a fascinating finding. Can you elaborate on the conditions or training data that you believe led to this behavior? Is it a general property of the Edit Flow text generation, or is it specific to the multimodal pretraining mixture used?

---

> ### Author Response · Authors · 2025-11-18
> **Response to Reviewer SeK2**
>
> Thank you for your thorough review. Below, we address each of your questions and concerns.
>
> ## Novelty over Edit Flows
> Our contribution extends Edit Flow by enabling concurrent interleaved generation with variable-length sequences, which requires novel interleaved time scheduling theory (Section 2.3). Independent time schedules create train-test mismatch since $t_\text{img} ≤ t_\text{text}$ at inference. Additionally, OneFlow is the first non-autoregressive multimodal model capable of variable-length generation, a fundamental limitation of all prior diffusion-based multimodal models.
>
> ## KV Cache
> The lack of KV caching is inherent to all bidirectional attention models, not specific to OneFlow. We explicitly acknowledge this limitation, whereas many papers omit this detail. As noted in Muddit (Section 4.5) and the survey on discrete diffusion models (Section VI.C), bidirectional attention prevents theoretically lossless caching since key-value pairs must be recomputed when attending to updated tokens.
>
> We compare against AR models with KV cache to present the strongest baseline. Against AR without KV cache or other non-autoregressive models, OneFlow would be faster. Developing efficient caching for bidirectional models is an active research area and important future work.
>
> ## Ablations on k-scheduler
> Thank you for raising this issue. We conducted ablations over different k-schedulers and will include these results in the camera-ready version. As described in DFM (Gat et al., 2024), there are three variants of k-scheduler: linear, quadratic, and cubic. We find that the linear scheduler works best.
>
> | K Scheduler Degree | CIDEr | ROUGE_L |
> | :---- | :---- | :---- |
> | 1.0 | 1.2925 | 0.5934 |
> | 2.0 | 1.2423 | 0.5821 |
> | 3.0 | 1.1173 | 0.5588 |
>
> ## t-Independent Parameterization
> Thank you for raising this issue. We empirically found t-independent parameterization to be superior despite theoretical motivation for t-dependence. This likely occurs because $X_t$ already encodes sufficient information about the noise level. We will include quantitative ablations in the revision. We ablated over the location of the time token insertion: begin_of_seq is the beginning of the sequence, begin_of_text is at the start of text tokens, and none is t-independent.
>
> | Time Location | CIDEr |
> | :---- | :---- |
> | begin_of_seq | 1.2776 |
> | begin_of_text | 0.3362 |
> | **none** | **1.2831** |
>
> ## Mixed Generation Probability
> Thank you for raising this issue. We ablated over different mixed-generation probabilities at 250k steps (note that the paper experiments use 500k steps). Using higher mixed-generation probability increases our CIDEr score from 131 to 135, showing a clear trend that higher mixed-generation probability improves image understanding. There is no clear trend for image generation, suggesting that mixed-generation probability does not hurt image generation capability while helping image understanding.
>
> | Mixed Gen Prob | CIDEr | ROUGE_L | Bleu_4 | FID |
> | :---- | :---- | :---- | :---- | :---- |
> | 0.10 | 129.86 | 0.5933 | 0.37 | 16.96 |
> | 0.20 | 131.40 | 0.5935 | 0.38 | 15.46 |
> | 0.40 | 132.80 | 0.5976 | 0.39 | 16.77 |
> | 0.80 | 135.41 | 0.6032 | 0.39 | 15.34 |
> | 1.00 | 135.65 | 0.6035 | 0.40 | 16.02 |
>
> ## Hallucination Analysis
> As shown below, CFG scales create a tradeoff between details and hallucinations. Higher CFG increases detail but also hallucinations, giving users control over this tradeoff.
>
> |  | CHAIRs (lower) | CHAIRi (lower) | Rating (higher) |
> | :---- | :---- | :---- | :---- |
> | AR | 3.0 | 2.3 | 5.60 |
> | **OneFlow** |  |  |  |
> | CFG 0.0 | 2.5 | 1.7 | 5.37 |
> | CFG 0.4 | 2.7 | 1.8 | 5.48 |
> | CFG 0.8 | 2.9 | 2.1 | 5.51 |
> | CFG 1.0 | 3.1 | 2.4 | 5.61 |
> | CFG 1.4 | 5.6 | 3.9 | 5.53 |
> | CFG 2.0 | 7.5 | 5.1 | 5.44 |
>
> ## Emergent Reasoning
> This is an inherent property of Edit Flow's hierarchical generation: it prioritizes content over grammar since insertions can occur anywhere, unlike AR's strict left-to-right generation requiring simultaneous grammar and content correctness. In Appendix F.5, we show that AR commits to incorrect answers early (saying "5" then changing to "3"), while OneFlow generates reasoning first and conclusions last. We include generation GIFs in supplementary materials demonstrating this process.
>
> ## Formatting Concerns
> Thank you for raising this issue, we will update the camera-ready version to address these concerns.
>
> Thank you for helping us improve our paper!
>
> ## References
> - Shi et al., 2025: "Muddit: Liberating generation beyond text-to-image with a unified discrete diffusion model" (Section 4.5)
> - Discrete Diffusion Survey: "Discrete Diffusion in Large Language and Multimodal Models: A Survey" (Section VI.C)
> - Gat et al., 2024: "Discrete flow matching"
> - Rohrbach, A., Hendricks, L.A., et al., 2018: "Object Hallucination in Image Captioning" (CHAIR metrics)

---

### Author Response · Authors · 2025-11-18
**General Response**

We thank all reviewers for their constructive feedback on inference speed, hallucination analysis, and scheduler ablations.

## KV Cache
The lack of KV caching is inherent to all bidirectional attention models, not specific to OneFlow. We explicitly acknowledge this limitation, whereas many papers omit this detail. As noted in Muddit (Section 4.5) and the survey on discrete diffusion models (Section VI.C), bidirectional attention prevents theoretically lossless caching since key-value pairs must be recomputed when attending to updated tokens.

**Computational Complexity Analysis:**
As shown by Muddit's inference complexity analysis:
- AR with KV Cache: O(L²D)
- AR without KV Cache: O(L³D) total FLOPs
- OneFlow and Discrete Diffusion Models: O(TL²D), where T ≪ L (typically 10-50 vs 512+ tokens)

Where L is the sequence length, D is the model dimension, and T is the number of sampling steps.

**Performance Trade-offs:**
While bidirectional attention prevents KV caching (a fundamental limitation shared by all discrete diffusion models), OneFlow compensates through substantially fewer sampling steps. We compare against AR with KV cache to present the strongest baseline. Our results show OneFlow achieves AR-level performance using only 6 sampling steps (Figure 16), demonstrating competitive wall-clock time despite the caching limitation. Against AR without KV cache or other non-autoregressive models, OneFlow would be significantly faster. Developing efficient caching mechanisms for bidirectional diffusion models remains an active research area and important future direction.

## Hallucination Analysis
CFG scaling creates a controllable precision-detail tradeoff. At lower CFG values (0.0-0.8), OneFlow maintains lower hallucination rates than AR (CHAIRs: 2.5-2.9 vs 3.0) while producing shorter captions (Rating: 5.37-5.51 vs 5.60). At CFG=1.0, OneFlow matches AR's caption detail (Rating: 5.61) with comparable hallucination rates (CHAIRs: 3.1 vs 3.0, CHAIRi: 2.4 vs 2.3). Higher CFG values (1.4-2.0) generate more detailed captions but increase hallucinations, providing users explicit control over the precision-detail tradeoff based on application requirements.

|  | CHAIRs (lower) | CHAIRi (lower) | Rating (higher) |
| :---- | :---- | :---- | :---- |
| AR | 3.0 | 2.3 | 5.60 |
| **OneFlow** |  |  |  |
| CFG 0.0 | 2.5 | 1.7 | 5.37 |
| CFG 0.4 | 2.7 | 1.8 | 5.48 |
| CFG 0.8 | 2.9 | 2.1 | 5.51 |
| CFG 1.0 | 3.1 | 2.4 | 5.61 |
| CFG 1.4 | 5.6 | 3.9 | 5.53 |
| CFG 2.0 | 7.5 | 5.1 | 5.44 |

## K Scheduler Ablations
We ablated over different k-scheduler variants as described in DFM (Gat et al., 2024): linear, quadratic, and cubic. Results demonstrate that the linear scheduler (degree 1.0) achieves the best performance across both CIDEr (129) and ROUGE_L (0.59) metrics. Performance degrades monotonically with higher polynomial degrees, with cubic scheduling showing substantial drops (CIDEr: 111, ROUGE_L: 0.55). This suggests that uniform token insertion rates are more effective than accelerated scheduling for our multimodal generation task. Full ablations will be included in the camera-ready version.

| K Scheduler Degree | CIDEr | ROUGE_L |
| :---- | :---- | :---- |
| 1.0 | 129.25 | 0.5934 |
| 2.0 | 124.23 | 0.5821 |
| 3.0 | 111.73 | 0.5588 |

## Paper Focus
Our paper's primary contribution is demonstrating that non-autoregressive generation can match autoregressive performance in multimodal settings through rigorous controlled experiments. We train our own AR baseline with identical data, architecture, and training procedures to enable fair apples-to-apples comparison, rather than comparing against external models with different training conditions. This controlled setup allows us to isolate the impact of the generation paradigm itself.

## References
- Shi et al., 2025: "Muddit: Liberating generation beyond text-to-image with a unified discrete diffusion model" (Section 4.5)
- Discrete Diffusion Survey: "Discrete Diffusion in Large Language and Multimodal Models: A Survey" (Section VI.C)
- Gat et al., 2024: "Discrete flow matching"

---

### Author Response · Authors · 2025-11-28
**Summary of Changes**

Thank you to all the reviewers for your feedback. We would like to summarize the discussion during the rebuttal period and the changes we made to the paper.

### KV Cache
The lack of KV caching is inherent to all bidirectional attention models, not specific to OneFlow. We explicitly acknowledge this limitation, whereas many papers omit this detail. As noted in Muddit (Section 4.5) and the survey on discrete diffusion models (Section VI.C), bidirectional attention prevents theoretically lossless caching since key-value pairs must be recomputed when attending to updated tokens.

### Hallucination Analysis
We conducted comprehensive hallucination analysis using CHAIR metrics across different CFG values (added to paper). Results show that CFG scaling creates a controllable precision-detail tradeoff: at CFG=1.0, OneFlow matches AR's caption detail with comparable hallucination rates (CHAIRs: 3.1 vs 3.0), while lower CFG values reduce hallucinations at the cost of shorter captions.

### New Ablation Studies
We added three ablation studies to the appendix:

1. Appendix F.4: t-independent parameterization ablation showing that removing explicit time conditioning (none) achieves the best performance (CIDEr: 128.31) compared to begin_of_seq (127.76) or begin_of_text (33.62).

2. Appendix F.5: k-scheduler ablation demonstrating that linear scheduling (degree 1.0) outperforms quadratic and cubic variants across both CIDEr and ROUGE_L metrics.

3. Appendix F.6: Mixed generation probability ablation showing that higher mixed-generation probability improves image understanding without hurting image generation performance.

### Expanded Comparisons
We added BAGEL, and Mogao to the VQA comparison table. We emphasize that our primary contribution is demonstrating non-autoregressive generation can match autoregressive performance through controlled experiments with identical training conditions.

### Animation of Generation Process
We include gifs of the generation process for a few VQA samples to demonstrate the reasoning process in the supplementary material.

---

### Meta-Review · Area_Chair_G2UP · 2026-01-07

**Summary:**

The AC carefully read the paper and the full discussion. The submission received mixed reviews (initial scores: 6, 6, 4, 6). Reviewers generally agreed that the paper effectively unifies recent diffusion-based text generation techniques with well-established diffusion-based image generation methods within a single transformer, and appreciated the rigorous experimental evaluation across a diverse set of benchmarks, metrics, and baselines. However, the main concerns focus on limited novelty—primarily framing the contribution as a straightforward combination of existing techniques for a new multimodal task—as well as high inference cost and the lack of comparisons to the latest state-of-the-art models (e.g., Mango and Show-o2). These issues raise questions about robustness, generality, and practical usability. As a result, it remains unclear whether the proposed method delivers clear, meaningful benefits for current or future unified multimodal models. Given that these core issues seem unlikely to be resolved through further discussion, I am inclined to recommend rejection.

**Reviewer Concerns:**

Some concerns regarding the missing ablations (Reviewer SeK2), the evaluation of interleaved and concurrent generation (Reviewer rS7v), and the previously unclear details of the interleaving setup (Reviewer wDyu) have been addressed. However, several non-negligible issues remain:

Reviewer rS7v highlighted the limited novelty. In particular, the main architectural contribution largely combines existing techniques for a new multimodal setting. As a result, the strong performance appears to rely heavily on the pre-established Edit Flows framework, and the additional methodological innovations beyond this integration seem incremental.

Reviewers SeK2 and qxm9 raised concerns about the high inference cost due to the inability to use KV caching. While this limitation is inherent to bidirectional attention models in general rather than specific to this work, the paper should include clearer comparisons against autoregressive models that benefit from caching, along with discussion or potential mitigation strategies, to better position the approach as a practical direction for future unified multimodal models.

Reviewer qxm9 also noted the lack of comparisons to more recent state-of-the-art models. This is a valid concern, especially given that models such as Show-o2 (NeurIPS 2025) have been released and open-sourced.

Overall, it remains unclear whether the proposed approach provides clear, meaningful benefits for current or future unified multimodal models.

**Reviewer Scores:**

Reviewer SeK2 is likely to keep the original score, mainly due to ongoing concerns about novelty.

Reviewer rS7v is expected to maintain the current score, since the minor issues raised have been addressed.

Reviewer qxm9 will likely retain the negative score, as the concerns about inference cost and missing comparisons to state-of-the-art models remain unresolved.

Reviewer wDyu is also likely to keep the original score.

---

### Decision · Program_Chairs · 2026-01-26

Reject